# StepCodeReasoner: Aligning Code Reasoning with Stepwise Execution Traces via Reinforcement Learning

Hao Wang[1]  Lei Sha[1]  Jie M. Zhang[2]

## Abstract

Existing code reasoning methods primarily supervise final code outputs, ignoring intermediate states, often leading to *reward hacking* where correct answers are obtained through inconsistent reasoning. We propose **StepCodeReasoner**, a framework that introduces *explicit intermediate execution-state supervision*. By automatically inserting structured `print`-based execution-trace anchors into code, the model is trained to predict runtime states at each step, transforming code reasoning into a verifiable, stepwise execution modeling problem. Building on this execution-aware method, we introduce **Dual-Granularity GRPO (DG-GRPO)**, a reinforcement learning algorithm for structured credit assignment at two complementary granularities: inter-trajectory comparison across sampled execution paths and intra-trajectory shaping based on downstream correctness. Extensive experiments demonstrate that StepCodeReasoner achieves state-of-the-art performance in code reasoning. In particular, our 7B model achieves 91.1% on CRUXEval and 86.5% on LiveCodeBench, outperforming the CodeReasoner-7B baseline (86.0% and 77.7%) and GPT-4o (85.6% and 75.1%). Furthermore, on the execution-trace benchmark REval, our model scores 82.9%, outperforming baseline CodeReasoner-7B (72.3%), its 14B counterpart (81.1%), and GPT-4o (77.3%). In addition, StepCodeReasoner improves code generation, achieving 90.1 on HumanEval, 85.0 on MBPP, and 19.4 on LiveCodeBench generation, with an average score of 64.8 versus 62.6 for CodeReasoner-7B.

[1]School of Artificial Intelligence, Beihang University, Beijing, China [2]King's College London, United Kingdom. Correspondence to: Jie M. Zhang <jie.zhang@kcl.ac.uk>, Lei Sha <shalei@buaa.edu.cn>.

*Proceedings of the 43rd International Conference on Machine Learning*, Seoul, South Korea. PMLR 306, 2026. Copyright 2026 by the author(s).

## 1. Introduction

Reasoning about code—predicting how a program evolves during execution—lies at the core of advanced Large Language Model (LLM) intelligence. Unlike surface-level pattern recognition, code reasoning requires models to faithfully simulate program semantics, including variable updates, control flow transitions, and long-range dependencies induced by loops and conditionals. This capability is essential for a wide range of software engineering tasks, such as automated debugging (Zhong et al., 2024), program repair (Ni et al., 2024; Ye et al., 2022), and algorithmic code generation.

Despite recent progress, most existing training paradigms still formulate code reasoning as a *terminal prediction* problem. Both supervised fine-tuning (SFT) and reinforcement learning (RL) approaches—including execution-centric methods such as CodeReasoner (Tang et al., 2025)—primarily optimize models based on the correctness of the final output. Intermediate reasoning steps, even when explicitly generated, are not verified against true program execution. As a result, models are free to arrive at correct answers through logically inconsistent or partially incorrect execution paths, a phenomenon often referred to as *reward hacking*. Because intermediate states are never observed or penalized, training signals fail to distinguish genuine execution understanding from lucky guesses.

To address this challenge, we propose **StepCodeReasoner**, a framework that transforms code reasoning from a blackbox prediction task into an explicit execution modeling problem with dense, structured supervision (as illustrated in Figure 1). The framework begins by making program execution states observable through the automatic insertion of **execution-trace anchors**. By implementing structured `print` statements at critical junctures—such as variable updates, loop exits, and return points—we convert abstract program logic into a series of concrete, verifiable targets. Building on these anchors, we decouple code reasoning into two distinct tasks: **output prediction** and **input prediction**. Unlike traditional methods that treat these as simple *assertion completion*, we employ specialized, execution-aligned prompts for each task. This unified yet decoupled format requires the model to interleave logical reasoning with state

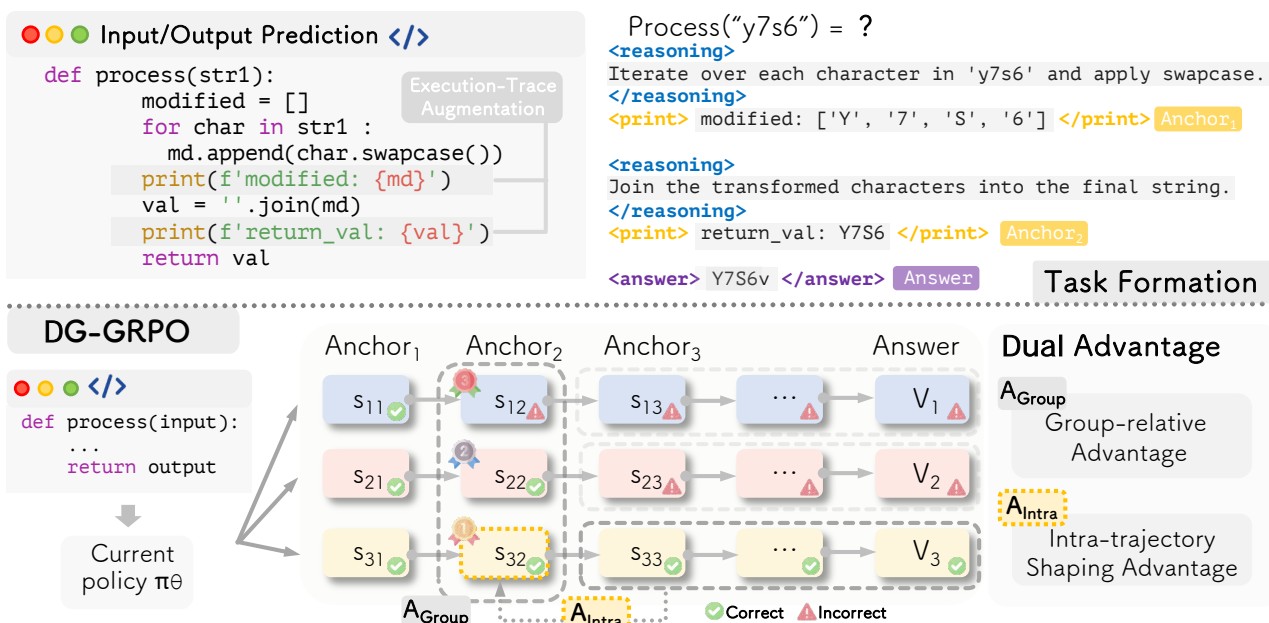

*Figure 1.* Overview of the StepCodeReasoner framework. The top-left illustrates **Execution-Trace Augmentation**, where the original `Swapcase` code is modified by inserting structured `print` statements to create observable execution anchors. The top-right shows the resulting **Model Inference Process** for the input `"y7s6"`, where the model generates interleaved blocks of `<reasoning>` for logical deduction and `<print>` for state prediction (e.g., $Anchor_1$ for the intermediate variable state and $Anchor_2$ for the returned string). The lower section details the Dual-Granularity GRPO training paradigm, where the current policy $\pi_\theta$ generates a group of trajectories for a single input, represented as a matrix of states $s_{i,j}$ and final values $v_i$. Notably, although $s_{32}$ and $s_{22}$ both represent correct intermediate steps, $s_{32}$ exhibits a higher advantage ($s_{32} > s_{22} > s_{12}$) because it belongs to a trajectory with higher subsequent rewards. The learning signal is refined through two distinct components: the Group-relative Advantage ($A_{Group}$) which contrasts the quality of different trajectories within the group to filter out noise, and the Intra-trajectory Shaping Advantage ($A_{Intra}$) which propagates rewards through sequential anchors within a single path. This dual-granularity advantage mechanism allows for precise credit assignment at each reasoning step, ultimately optimizing the model's ability to predict both intermediate execution states and final outputs correctly.

prediction, ensuring its deductions are grounded in program semantics.

To provide rigorous supervision for these reasoning chains, we introduce a novel reinforcement learning algorithm, **Dual-Granularity GRPO (DG-GRPO)**, which provides fine-grained supervision over both intermediate execution steps and final outcomes. Unlike standard RL methods that assign a single sparse reward at the end of a trajectory, our approach computes stepwise rewards at each execution anchor and performs group-relative normalization across sampled trajectories. Crucially, DG-GRPO further incorporates an *intra-trajectory shaping signal* that modulates step-level credit based on how well a correct intermediate prediction facilitates subsequent correct execution. We use "dual-granularity" to emphasize that credit assignment operates across trajectories and within trajectories, without implying an outer–inner optimization problem.

Extensive experiments demonstrate that **StepCodeReasoner** achieves a new state-of-the-art in code reasoning. Compared to the leading open-source baseline CodeReasoner-7B (81.8% average), our 7B model delivers an average score of 87.8% across prediction tasks and scores 82.9% on the fine-grained REval benchmark, surpassing the baseline's 72.3%. Notably, our model outperforms the proprietary GPT-4o (80.3% average), demonstrating superior performance on the input prediction tasks of Live-CodeBench and CRUXEval. These results suggest that learning to explicitly execute code provides a far more robust foundation for reasoning than merely predicting final outputs.

In summary, this work makes the following contributions:

- We propose **StepCodeReasoner**, a framework that introduces execution anchors to expose intermediate program states and designs distinct, execution-aligned prompting schemes for input prediction and output prediction in code reasoning.

- We develop **Dual-Granularity GRPO (DG-GRPO)**, a reinforcement learning algorithm that performs structured credit assignment at two complementary granularities: across trajectory groups via relative comparison, and within individual trajectories to model long-range

execution dependencies.

- We demonstrate that **StepCodeReasoner** allows our 7B model to surpass GPT-4o on multiple code reasoning benchmarks (CRUXEval, REval) and significantly improves performance on algorithmic code generation tasks like LiveCodeBench, showing that learning to execute code improves the ability to write code.

## 2. Related Work

### 2.1. Code Reasoning

Code reasoning is commonly evaluated through execution-oriented benchmarks that require models to infer program behavior from code, such as predicting inputs or outputs under partial execution settings (Gu et al., 2024; Jain et al., 2024). Follow-up work extends this paradigm by incorporating natural language specifications (Chen et al., 2024), supporting multiple programming languages (Xu et al., 2024), or constructing reasoning tasks from real-world software projects (Roy et al., 2025), enabling broader analysis of model generalization across settings.

Beyond evaluation, recent work has explored training strategies to improve models' code reasoning abilities. Code-Executor and TRACED incorporate execution supervision or traces during pre-training to capture dynamic program behavior (Liu et al., 2023; Ding et al., 2024b), while Sem-Coder further reasons over comprehensive execution semantics using natural language monologue reasoning (Ding et al., 2024a). Other approaches use code reasoning tasks to enhance broader reasoning ability, including CODEI/O through input–output prediction with natural language rationales (Li et al., 2025a) and Absolute Zero via self-play reinforcement learning on executable tasks (Zhao et al., 2025). In contrast, CodeReasoner directly optimizes code reasoning performance through execution-focused data construction, supervised fine-tuning, and reinforcement learning (Tang et al., 2025). However, most existing methods primarily supervise final outcomes or use traces as data augmentation, whereas our approach uses interpreter-verified anchors as step-level RL rewards.

### 2.2. Reinforcement Learning in LLMs for Reasoning

Recent advances in reinforcement learning have substantially improved the reasoning capabilities of large language models. Group Relative Policy Optimization (GRPO) (Shao et al., 2024) has become a widely adopted foundation, with subsequent work focusing on reducing variance and improving training stability. DAPO (Yu et al., 2025) and ASPO (Wang et al., 2025) mitigate optimization bias and instability through dynamic sampling strategies and asymmetric importance weighting, respectively. From an optimization perspective, GSPO (Zheng et al., 2025) argues

that token-level importance ratios introduce unnecessary variance, and instead performs policy optimization using sequence-level likelihood ratios and advantages. Complementary to this coarse-grained view, Step-GRPO (R1-VL) (Zhang et al., 2025) refines GRPO with step-wise relative advantage estimation, decomposing long-chain reasoning into intermediate steps and assigning fine-grained credit using rule-based rewards derived from ground-truth reasoning paths. Beyond flat rollouts, TreePO (Li et al., 2025b) further restructures exploration into a tree search, exploiting hierarchical relative advantages among sibling nodes to improve both exploration efficiency and policy learning.

Reinforcement learning has also been widely applied in software engineering tasks that require structured decision-making over long horizons. In code generation and completion, RL is used to optimize retrieval-augmented pipelines for selecting relevant repository context (Wang et al., 2024) and to train critic models that evaluate partial code during generation (Li et al., 2024). For automated program repair, RL agents explore the patch search space while balancing functional correctness and similarity to reference implementations (Wei et al., 2025). RL has also shown strong performance in software testing, where it guides fuzzers toward inputs that maximize coverage and vulnerability discovery (Li et al., 2022; Eom et al., 2024). Collectively, these works highlight RL as a unifying framework for advancing code reasoning across generation, repair, and verification.

## 3. Method

### 3.1. Problem Definition

Before detailing our specific framework, we first formalize the code reasoning tasks and contrast the traditional terminal supervision paradigm with our proposed stepwise objective. Code reasoning tasks can be categorized into two types: **output prediction**, where the model predicts the result $Y$ given a program $P$ and input $I$; and **input prediction**, where the model identifies a valid input $I$ that produces a target output $Y$. For clarity, we use $C$ to represent the known condition and $V$ to represent the missing target value to be predicted.

Input prediction does not assume that the program is injective. For a target output $Y^*$, there may be multiple valid inputs $\mathcal{I}^*(P, Y^*) = \{I : \mathcal{E}(P, I) = Y^*\}$. We use one valid input for SFT demonstrations, while RL and evaluation accept any predicted input $\hat{I}$ satisfying $\mathcal{E}(P, \hat{I}) = Y^*$.

**Why Learn Execution Reasoning?** When a complete executable program and concrete input are available, a Python interpreter is the correct tool for obtaining the exact output. Our goal is not to replace the interpreter, but to use it as a supervision oracle so the model can internalize execution

semantics for settings such as unknown-input reasoning, partial or generated code, debugging, repair, and code generation.

**The Final-Answer–Only Approach**   Under the traditional terminal supervision paradigm, code reasoning is formulated as a single-step prediction problem. Given the program $P$ and the condition $C$, the model is trained to directly produce the target value $V$. Although the model may generate an intermediate reasoning trace, the training process only supervises the final answer. Formally, the optimization objective is defined as:

$$\mathcal{L}_{\text{terminal}}(\theta) = -\log p_\theta(V^* \mid P, C) \tag{1}$$

This objective treats the execution process as an unobserved process. Consequently, any reasoning path that leads to the correct answer $V^*$ is rewarded equally, even if the intermediate steps are logically inconsistent with the actual program behavior. This often leads to "right answer, wrong logic," where the model relies on superficial patterns rather than true simulation.

**The StepCodeReasoner Approach**   In contrast, our StepCodeReasoner framework reformulates code reasoning as a stepwise execution modeling task. We make the internal execution visible by inserting a sequence of anchors into $P$, which act as checkpoints for the program's state. The model is now required to predict the complete execution trajectory $S^* = \{s_1^*, s_2^*, \ldots, s_n^*\}$, where each $s_i^*$ represents the true values of variables at the $i$-th anchor, followed by the final value $V^*$. The optimization objective is thus defined as:

$$\mathcal{L}_{\text{StepCodeReasoner}}(\theta) = -\sum_{i=1}^{n+1} \log p_\theta(z_i^* \mid P, C, z_{<i}^*), \tag{2}$$

where

$$z_i = \begin{cases} s_i, & i \le n, \\ V, & i = n+1. \end{cases} \tag{3}$$

In this formulation, every intermediate state $s_i^*$ is explicitly supervised and verified against the ground truth from a code interpreter. Importantly, all ground-truth supervision—both intermediate states and final outputs—is generated by a deterministic Python interpreter, not by the LLM itself. The model's role is to learn to *predict* interpreter output, thereby internalizing execution semantics as a reasoning capability that can be deployed without a sandbox at inference time. By assigning rewards to each step, StepCodeReasoner encourages reasoning paths to align with the program's actual execution logic, substantially reducing the chance of obtaining a correct final answer through flawed intermediate reasoning.

## 3.2. Execution-Trace Augmentation and Task Decoupling

Having established the stepwise supervision objective, we now describe the data construction pipeline required to instantiate this process, specifically how we generate and structure the execution traces to make internal logic visible. The effectiveness of StepCodeReasoner relies on the quality and density of the execution-trace anchors. To transform a raw program $P$ into an instrumented version $P'$ that surfaces its internal states, we employ a teacher LLM to automatically insert `print()` statements. This process is governed by a set of heuristic rules designed to maximize logical transparency while maintaining a manageable sequence length. We define a transformation function $\mathcal{T}$ that performs this mapping:

$$P' = \mathcal{T}(P) \tag{4}$$

Unlike naive instrumentation that might trace every line, we implement a selective strategy to avoid trace explosion. Specifically, we prohibit the insertion of anchors inside `for` or `while` loops, as repetitive state logging can lead to excessively long sequences that exceed the model's context window. Instead, the state of variables affected by a loop is logged immediately after the loop terminates. Furthermore, we mandate anchors after significant variable assignments and immediately before a function's `return` statement. To ensure the states are easily parsable, all anchors follow a strict format: `print(f'VAR_NAME: {VAR_NAME}')`. This instrumentation process is non-destructive; the teacher model is permitted to unroll complex one-liners into multi-line blocks only when necessary to insert a checkpoint.

The teacher LLM only proposes where to expose states; it does not provide the supervision values. After instrumentation, every candidate program is executed by a deterministic interpreter, and programs that fail to execute or produce inconsistent behavior are discarded. This design limits the effect of teacher artifacts: suboptimal anchor placement can reduce supervision density, but the surviving anchor values remain interpreter-generated rather than LLM-generated. When $P'$ is executed by a deterministic interpreter $\mathcal{E}$ with input $I$, it yields a sequence of ground-truth intermediate states $S^*$:

$$S^* = \{s_1^*, s_2^*, \ldots, s_n^*\} = \mathcal{E}(P', I) \tag{5}$$

To bridge the gap between code execution and natural language reasoning, we define structured templates that decouple the reasoning process into verifiable segments. For the standard **output prediction** task, the model is required to follow an iterative *Reason-then-Print* cycle. For every anchor in the augmented code $P'$, the model must first generate a `<reasoning>` block $r_i$ to simulate the state transition, followed by a `<print>` block $s_i$ containing the

predicted value. After all anchors are predicted, a final deduction leads to the answer $V$. We represent the model's generated sequence $\mathbf{y}_{\text{output}}$ as follows:

$$\mathbf{y}_{\text{output}} = (r_1, s_1) \oplus (r_2, s_2) \oplus \cdots \oplus (r_n, s_n) \oplus (r_V, V) \quad (6)$$

where $\oplus$ denotes string concatenation. This structure ensures that the final answer is a direct consequence of the step-by-step simulation.

For the more complex **input prediction** task, we introduce an additional **input inference module** at the beginning of the sequence. Before starting the execution simulation, the model must first use a `<reasoning>` block to analyze the code's requirements and determine an unknown input $I$. This input is then provided in an `<input>` tag, acting as a "commitment" for the rest of the trace. Only after this input is fixed does the model proceed with the standard *Reason-then-Print* cycle to verify the execution flow. The sequence $\mathbf{y}_{\text{input}}$ is defined as:

$$\mathbf{y}_{\text{input}} = \underbrace{(r_I, I)}_{\text{Input Inference}} \oplus \underbrace{(r_1, s_1) \oplus \cdots \oplus (r_n, s_n)}_{\text{Trace Verification}} \oplus \underbrace{(r_V, V)}_{\text{Final Deduction}}$$
(7)

At inference time, this trace is a consistency check for the committed input rather than an iterative search procedure. During RL training, however, incorrect trace states receive zero stepwise rewards even when the final answer is plausible, encouraging inputs that yield globally consistent execution trajectories. By forcing the model to explicitly predict $I$, $s_i$, and $V$ in separate, labeled blocks, we transform code reasoning into a transparent chain of verifiable claims.

### 3.3. Dual-Granularity GRPO Optimization

To exploit dense execution-level supervision while explicitly modeling structured dependencies within each sampled execution trajectory, we propose **Dual-Granularity GRPO (DG-GRPO)**, a group-relative policy optimization algorithm with intra-trajectory credit modulation. Unlike stepwise methods such as Step-GRPO, which estimate advantages solely via cross-trajectory comparisons at the same execution anchor, DG-GRPO augments group-relative optimization with a trajectory-aware shaping signal that reflects how individual steps facilitate subsequent correct execution.

For a given problem context $x$, we sample a group of $G$ trajectories $\{\mathbf{y}_1, \ldots, \mathbf{y}_G\}$ from the current policy $\pi_\theta$. Each trajectory $\mathbf{y}_g$ is represented as a sequence of predicted execution states followed by a final output:

$$\mathbf{y}_g = \{s_{1,g}, s_{2,g}, \ldots, s_{n,g}, V_g\},$$

where $s_{i,g}$ denotes the model-predicted runtime state at the $i$-th execution anchor, $V_g$ is the predicted final answer, and $n$ is the total number of verifiable execution anchors. Importantly, $n$ is determined by the instrumented program $P'$,

not by the model generation. Therefore, all $G$ trajectories sampled for the same problem context share the same ordered anchor set, making the group-relative comparison at each anchor index well-defined. If a response omits an anchor because of formatting errors or early truncation, we assign zero reward to the missing anchor and all subsequent predictions in that trajectory.

**Execution-Level Rewards.** We define a multi-level reward function that supervises both intermediate execution states and the final output. For each execution anchor $i$, a binary stepwise reward is defined as

$$r_{i,g} = \begin{cases} 1, & \text{if } s_{i,g} = s_i^* \\ 0, & \text{otherwise,} \end{cases} \quad (8)$$

where $s_i^*$ is the ground-truth runtime state obtained by executing the target program. The terminal reward is defined as

$$r_{\text{final},g} = \begin{cases} 1, & \text{if } V_g = V^* \\ 0, & \text{otherwise,} \end{cases} \quad (9)$$

where $V^*$ denotes the ground-truth program output.

**Group-Relative Stepwise Advantage.** For each anchor $i$, we compute a group-relative advantage by normalizing stepwise rewards across the sampled group:

$$\hat{A}_{i,g}^{\text{group}} = \frac{r_{i,g} - \frac{1}{G} \sum_{g'=1}^{G} r_{i,g'}}{\sqrt{\frac{1}{G} \sum_{g'=1}^{G} \left( r_{i,g'} - \frac{1}{G} \sum_{g''=1}^{G} r_{i,g''} \right)^2 + \epsilon}},$$
(10)

where $G$ is the group size and $\epsilon$ is a small constant for numerical stability. This term encourages learning from trajectories that outperform peers at the same execution anchor. This anchor-aligned normalization differs from pooled step-level normalization in TANGO (Zha et al., 2025): because our trajectories share deterministic execution anchors, the $i$-th prediction is compared only with predictions for the same program checkpoint.

**Intra-Trajectory Shaping Advantage.** While group-relative normalization captures cross-trajectory competitiveness, it does not distinguish steps that merely achieve local correctness from those that meaningfully enable subsequent correct execution. To address this, we introduce an intra-trajectory shaping advantage that modulates step-level credit based on future execution outcomes.

For each non-terminal anchor $i < n$, we first compute the empirical future success rate

$$F_{i,g} = \frac{1}{n-i} \sum_{j=i+1}^{n} r_{j,g}. \quad (11)$$

The intra-trajectory shaping advantage is then defined as

$$\hat{A}_{i,g}^{\text{intra}} = \begin{cases} r_{i,g} \cdot (1 + F_{i,g}), & i < n, \\ r_{n,g}, & i = n, \end{cases} \quad (12)$$

where the future-correctness term is zero for the terminal anchor. This formulation has three key properties: (i) only correct steps are reinforced; (ii) correct steps followed by more correct future execution receive proportionally higher credit; and (iii) the formulation requires no additional value function or discount hyperparameters.

Although this term is not a standard return estimator such as GAE (Schulman et al., 2015), it has a reward-shaping interpretation: $F_{i,g}$ estimates future success under binary rewards, so correct steps receive more credit when the remaining trajectory also stays correct. This is analogous in spirit to potential-based reward shaping (Ng et al., 1999), while remaining critic-free and directly computable from interpreter-verified anchors.

**Dual-Granularity Stepwise Advantage.** We combine group-relative and intra-trajectory advantages additively:

$$\hat{A}_{i,g} = \hat{A}_{i,g}^{\text{group}} + \lambda \cdot \hat{A}_{i,g}^{\text{intra}}, \quad (13)$$

where $\lambda > 0$ controls the relative contribution of intra-trajectory shaping. In this formulation, $\hat{A}^{\text{group}}$ serves as the primary policy gradient signal, while $\hat{A}^{\text{intra}}$ acts as an auxiliary shaping term that biases learning toward steps that facilitate successful future execution.

**Optimization Objective.** The stepwise optimization objective is defined as

$$\mathcal{L}_{\text{step}} = -\frac{1}{G} \sum_{g=1}^{G} \sum_{i=1}^{n} \frac{\pi_\theta(s_{i,g} \mid s_{<i,g}, x)}{\pi_{\theta_{\text{old}}}(s_{i,g} \mid s_{<i,g}, x)} \hat{A}_{i,g}, \quad (14)$$

where $\pi_{\theta_{\text{old}}}$ denotes the policy used to generate the sampled trajectories.

For the final output, we compute a group-relative advantage $\hat{A}_{\text{final},g}$ by normalizing $\{r_{\text{final},1}, \dots, r_{\text{final},G}\}$ analogously, and define the terminal optimization objective as

$$\mathcal{L}_{\text{final}} = -\frac{1}{G} \sum_{g=1}^{G} \frac{\pi_\theta(\mathbf{y}_g \mid x)}{\pi_{\theta_{\text{old}}}(\mathbf{y}_g \mid x)} \hat{A}_{\text{final},g}. \quad (15)$$

The overall training objective of DG-GRPO is

$$\mathcal{L}_{\text{DG-GRPO}} = \mathcal{L}_{\text{step}} + \mathcal{L}_{\text{final}} + \mathbb{D}_{\text{KL}}(\pi_\theta \parallel \pi_{\text{ref}}), \quad (16)$$

where $\pi_{\text{ref}}$ is a fixed reference policy and $\mathbb{D}_{\text{KL}}$ denotes the Kullback–Leibler divergence. By integrating group-relative optimization with trajectory-aware shaping, DG-GRPO enables fine-grained and structure-aligned credit assignment consistent with program execution semantics.

# 4. Experiments

## 4.1. Benchmarks and Evaluation Protocol

To evaluate **StepCodeReasoner**, we use three complementary benchmarks that test different levels of code execution reasoning. **CRUXEval** (Gu et al., 2024) evaluates basic function-level execution reasoning on short Python programs. It contains output prediction (CX-O), where the model predicts the output given a program and input, and input prediction (CX-I), where the model infers an input that produces a target output. **LiveCodeBench** (Jain et al., 2024) uses harder, contamination-free programming-contest problems and evaluates the same two capabilities (LCB-O and LCB-I), allowing us to test whether trace-based supervision scales beyond short benchmark functions to algorithm-level programs involving more complex control flow and data structures. **REval** (Chen et al., 2024) evaluates fine-grained execution understanding through four line-sensitive tasks: code coverage prediction (whether a line is executed), program state prediction (variable values at a target line), execution path prediction (the next line to be executed), and output prediction.

We instrument all benchmarks using the execution-trace augmentation procedure in Section 3.2. For CRUXEval and LiveCodeBench, the model is prompted to predict the outputs of the injected `print` anchors before producing the final answer. To ensure fair comparison with baselines that do not generate intermediate traces, the reported *pass@1* score depends only on the final answer inside the `<answer>` tag; intermediate traces are used to structure the reasoning process but are not counted in the final-answer metric. For REval, inserting `print` statements changes source-code line numbers, while the original tasks query specific lines. We therefore maintain an automated mapping $f : L_{\text{orig}} \to L_{\text{instr}}$ from original to instrumented line numbers and update all coverage, state, and path queries accordingly. REval performance is reported as accuracy on these updated fine-grained tasks.

Table 1 summarizes the distribution of execution-trace anchors after instrumentation. CRUXEval has fewer anchors because its functions are short, whereas LiveCodeBench yields denser traces due to competition-level algorithmic structure. During training, samples whose execution trace exceeds 10 lines are filtered to control context length. All models are evaluated using greedy decoding ($T = 0.0$). Appendix C provides the full prompt and dataset-specific processing details.

## 4.2. Dataset Construction

Our training data is primarily synthesized and adapted from the datasets provided by **CodeReasoner** (Tang et al., 2025). We enhance these raw code samples using our LLM-based

Table 1. Distribution of print anchors per sample across benchmarks.

| Benchmark | Mean | Median | Range |
|---|---|---|---|
| CRUXEval | 3.2 | 3 | 1–7 |
| LiveCodeBench | 4.8 | 5 | 2–10 |
| REval | 3.5 | 3 | 1–8 |
| Training Set | 4.1 | 4 | 1–10 |

instrumentor to generate augmented programs $P'$. During the ground-truth generation phase, we execute the augmented programs using a Python interpreter and apply execution validation and trace-length filtering to ensure data quality. For instances where the executed program yields only a single line of output ($n = 1$), it implies that the anchor was only placed at the final return statement; consequently, stepwise prediction in these cases becomes equivalent to terminal output prediction, and these samples are trained using only terminal-based reinforcement learning. To maintain computational efficiency and prevent the context window from being overwhelmed by excessively long reasoning chains, we filter out any data samples where the execution trace exceeds 10 lines of output. After this refinement and a rigorous decontamination process—following the 10-gram filtering method—we obtain 17,332 cases for supervised fine-tuning (SFT) and 18,796 cases for reinforcement learning. Appendix G and Appendix I provide robustness and preliminary scaling analyses.

### 4.3. Implementation Details

We adopt **Qwen2.5-Coder-7B-Instruct** as our primary base model. We use **GPT-4o** as the teacher model for synthesizing the initial reasoning paths. For the reinforcement learning phase, we utilize the GRPO algorithm via the `verl` framework. We set the total intermediate reward budget $R_{internal} = 1.0$ and the final answer reward $R_{final} = 1.0$. For the advantage formulation in Eq. (13), we set the weighting coefficient $\lambda$ to 0.3 in all experiments. We train for two epochs with a learning rate of 1e-6, sampling five candidate responses per prompt with a maximum length of 4,096 tokens. All experiments are conducted on a machine equipped with eight NVIDIA Tesla A100 GPUs, each with 40 GB of memory.

### 4.4. Baselines

We compare **StepCodeReasoner** with proprietary models (**GPT-4o** (OpenAI, 2024a), **GPT-4o-mini** (OpenAI, 2024)), strong open-source models (**Qwen2.5-72B-Instruct** (Team, 2024), **Llama3-70B-Instruct** (Dubey et al., 2024), **Qwen2.5-Coder-32B-Instruct** (Hui et al., 2024), **Qwen3-Coder-30B-A3B-Instruct** (Yang et al., 2025), **Qwen3.5-9B**), and code-reasoning methods representing

SFT-only, large-scale-data, and RL-based paradigms: **SEMCODER** (Ding et al., 2024a), **CODEI/O** (Li et al., 2025a), and **CodeReasoner** (Tang et al., 2025). Appendix H additionally compares our deterministic verifier with learned PRMs. All comparisons use our **7B StepCodeReasoner** model and greedy decoding ($T = 0.0$).

### 4.5. Main Results

#### 4.5.1. BASELINE RESULTS

Table 2. The performance comparisons between methods in input-output prediction and output-input prediction. Improvement rates for StepCodeReasoner are reported relative to CodeReasoner-7B.

| Model | Size | CRUXEval | | | LiveCodeBench | | | Avg |
|---|---|---|---|---|---|---|---|---|
| | | CX-O | CX-I | Avg | LCB-O | LCB-I | Avg | |
| GPT-4o | – | 0.905 | 0.806 | 0.856 | 0.848 | 0.653 | 0.751 | 0.803 |
| GPT-4o-mini | – | 0.769 | 0.673 | 0.721 | 0.777 | 0.591 | 0.684 | 0.703 |
| Qwen2.5 | 72B | 0.795 | 0.746 | 0.771 | 0.827 | 0.695 | 0.761 | 0.766 |
| Llama 3 | 70B | 0.637 | 0.613 | 0.625 | 0.564 | 0.526 | 0.545 | 0.585 |
| Qwen2.5-Coder | 32B | 0.752 | 0.834 | 0.793 | 0.806 | 0.678 | 0.742 | 0.768 |
| Qwen3-Coder | 30B | 0.778 | 0.861 | 0.820 | 0.832 | 0.706 | 0.769 | 0.794 |
| Qwen3.5 | 9B | 0.823 | 0.787 | 0.805 | 0.785 | 0.725 | 0.755 | 0.780 |
| SEMCODER | 6.7B | 0.625 | 0.651 | 0.638 | 0.597 | 0.530 | 0.564 | 0.601 |
| CODE I/O | 7B | 0.625 | 0.679 | 0.652 | 0.608 | 0.552 | 0.580 | 0.616 |
| CodeReasoner | 7B | 0.856 | 0.868 | 0.860 | 0.810 | 0.743 | 0.777 | 0.818 |
| CodeReasoner | 14B | 0.912 | 0.868 | 0.890 | 0.866 | 0.825 | 0.846 | 0.868 |
| **StepCodeReasoner** | **7B** | **0.916** | **0.905** | **0.911** | **0.884** | **0.845** | **0.865** | **0.878** |
| | | (+7.0%) | (+4.9%) | (+5.9%) | (+9.1%) | (+13.7%) | (+11.3%) | (+7.3%) |

As shown in Table 2, **StepCodeReasoner (7B)** achieves the best overall performance across all benchmarks, with the highest average score of **0.878**. Despite its relatively small model size, it consistently outperforms large open-source models (e.g., Qwen2.5-72B and Llama3-70B) as well as strong proprietary models such as GPT-4o, demonstrating strong parameter efficiency.

A notable observation is that StepCodeReasoner yields larger improvements on the input prediction tasks (CXEval-I and LCB-I) than on the corresponding output prediction tasks. This result indicates that decoupling input and output prediction, together with stepwise supervision over intermediate execution states, is particularly effective for improving input prediction, which relies more heavily on accurate intermediate reasoning.

Compared with code-reasoning-specific baselines including SEMCODER, CODEI/O, and CodeReasoner, StepCodeReasoner shows consistent gains across all settings. Notably, the 7B StepCodeReasoner substantially outperforms the 7B CodeReasoner and remains competitive with its 14B variant, highlighting the effectiveness of the proposed stepwise reinforcement learning framework.

#### 4.5.2. PERFORMANCE ON REVAL

Table 3 presents a detailed comparison on fine-grained code reasoning tasks, where StepCodeReasoner (7B) exhibits strong and well-balanced performance across all evaluation

*Table 3.* The performance comparisons between methods in REval. Improvement rates for StepCodeReasoner are reported relative to CodeReasoner-7B.

| Model | Size | Coverage | State | Path | Output | Avg |
|---|---|---|---|---|---|---|
| GPT-4o | – | 0.875 | 0.724 | 0.647 | 0.845 | 0.773 |
| GPT-4o-mini | – | 0.636 | 0.665 | 0.587 | 0.770 | 0.636 |
| Qwen2.5 | 72B | 0.885 | 0.699 | 0.601 | 0.836 | 0.755 |
| Llama 3 | 70B | 0.853 | 0.592 | 0.403 | 0.746 | 0.649 |
| Qwen2.5-Coder | 32B | 0.856 | 0.667 | **0.687** | 0.833 | 0.761 |
| Qwen3-Coder | 30B | 0.862 | 0.675 | 0.662 | 0.841 | 0.768 |
| SEMCODER | 6.7B | 0.467 | – | – | 0.562 | – |
| CODEI/O | 7B | 0.709 | 0.485 | 0.409 | 0.604 | 0.552 |
| CodeReasoner | 7B | 0.864 | 0.672 | 0.514 | 0.843 | 0.723 |
| CodeReasoner | 14B | 0.937 | 0.799 | 0.606 | 0.910 | 0.811 |
| **StepCodeReasoner** | **7B** | **0.944** | **0.823** | 0.631 | **0.918** | **0.829** |
| | | (+9.3%) | (+22.5%) | (+22.8%) | (+8.9%) | (+14.7%) |

dimensions. Despite its moderate scale, StepCodeReasoner achieves the highest average accuracy among all models, surpassing not only similarly sized baselines but also larger variants of CodeReasoner.

A closer inspection reveals that the most pronounced gains appear in *State* and *Path* prediction, which require accurate tracking of intermediate execution states and control-flow transitions. These aspects are explicitly targeted by the stepwise training objective, indicating that fine-grained intermediate supervision plays a critical role in strengthening execution-level reasoning. In contrast, improvements on Coverage and Output prediction are more moderate, suggesting that these tasks are less sensitive to step-level credit assignment.

Compared to proprietary and large open-source models, StepCodeReasoner maintains a favorable performance–efficiency trade-off. While larger models occasionally excel in isolated sub-tasks, such as Path prediction, they fail to achieve consistent gains across all reasoning dimensions. Notably, StepCodeReasoner-7B slightly outperforms CodeReasoner-14B on average, demonstrating that structured stepwise optimization can effectively compensate for, and even outweigh, differences in parameter scale.

Overall, these results confirm that explicitly modeling intermediate reasoning steps leads to more reliable fine-grained code understanding, enabling smaller models to compete with and surpass significantly larger alternatives on complex reasoning benchmarks.

### 4.5.3. PERFORMANCE ON CODE GENERATION

Although StepCodeReasoner is designed to improve stepwise code reasoning, it is important to assess whether such gains generalize to standard code generation tasks that evaluate only final program correctness. We therefore evaluate StepCodeReasoner on three widely used benchmarks—HumanEval (Chen et al., 2021), MBPP (Austin et al., 2021), and LiveCodeBench(v5) (Jain et al., 2024)—covering curated, large-scale, and real-world cod-

ing scenarios. HumanEval and MBPP evaluate functional correctness on manually curated and crowd-sourced Python problems, while LiveCodeBench(v5) provides a large-scale, contamination-free evaluation using newly released problems from competitive programming platforms. Together, these benchmarks allow us to examine whether execution-aware reasoning improvements translate into robust end-to-end code generation performance.

*Table 4.* Performance comparison on code generation benchmarks. Improvement rates for StepCodeReasoner are reported relative to CodeReasoner-7B.

| Model | Size | Code Generation Benchmarks | | | Avg |
|---|---|---|---|---|---|
| | | HumanEval | MBPP | LiveCodeBench | |
| Qwen2.5 | 72B | 86.0 | 80.2 | **20.4** | 62.2 |
| Llama 3 | 70B | 80.5 | 84.2 | 16.5 | 60.4 |
| Qwen2.5-Coder | 7B | 88.4 | 83.5 | 18.2 | 63.4 |
| SEMCODER | 7B | 79.3 | 79.6 | 16.0 | 58.3 |
| CodeReasoner | 7B | 87.6 | 82.4 | 17.8 | 62.6 |
| CodeReasoner | 14B | 88.9 | 84.1 | 18.6 | 63.9 |
| SFT + RL (Terminal Reward Only) | 7B | 88.2 | 83.1 | 18.1 | 63.1 |
| **StepCodeReasoner** | **7B** | **90.1** | **85.0** | 19.4 | **64.8** |
| | | (+2.9%) | (+3.2%) | (+9.0%) | (+3.5%) |

Table 4 shows that StepCodeReasoner also improves standard code generation. Compared with the same-budget terminal-only RL baseline, it performs better on HumanEval (90.1 vs. 88.2), MBPP (85.0 vs. 83.1), and LiveCodeBench generation (19.4 vs. 18.1), suggesting that the gains come from stepwise supervision rather than training budget alone.

### 4.5.4. ABLATION STUDY

To analyze the contribution of each component in StepCodeReasoner, we conduct ablation experiments by selectively modifying the training objectives and supervision signals.

- **SFT Only.** Train with supervised fine-tuning on both intermediate traces and final outputs, without RL. Evaluates if imitation alone suffices for faithful code reasoning, without leveraging any reinforcement learning signals, which might limit the model's ability to generalize to unseen tasks.

- **StepCodeReasoner Only (Ours, w/o SFT).** Train solely with stepwise trace-aligned RL, no supervised pretraining. Tests whether stepwise RL can recover correct execution dynamics from sparse initial behavior, without the benefit of supervised fine-tuning.

- **SFT + RL with Terminal Reward Only.** Apply SFT first, then RL using only final-output rewards. Serves as a conventional RL baseline to measure the benefit of stepwise rewards, focusing on the terminal reward without intermediate supervision, which may not fully capture the nuances of intermediate execution steps.

*Table 5.* Performance comparison of StepCodeReasoner and its ablations on CRUXEval, LCB-O, LCB-I and REval

| Model | CRUXEval | | LiveCodeBench | | REval | | | | Avg |
|---|---|---|---|---|---|---|---|---|---|
| | CXEval-O | CXEval-I | LCB-O | LCB-I | Coverage | State | Path | Output | |
| SFT Only | 0.770 | 0.750 | 0.730 | 0.700 | 0.742 | 0.582 | 0.421 | 0.703 | 0.700 |
| StepCodeReasoner Only (w/o SFT) | 0.820 | 0.805 | 0.790 | 0.760 | 0.781 | 0.641 | 0.487 | 0.746 | 0.740 |
| SFT + RL (Terminal Reward Only) | 0.880 | 0.865 | 0.845 | 0.810 | 0.823 | 0.691 | 0.532 | 0.801 | 0.777 |
| SFT + StepCodeReasoner (w/o Decoupling) | 0.900 | 0.888 | 0.860 | 0.825 | 0.902 | 0.818 | 0.625 | 0.855 | 0.835 |
| StepCodeReasoner (Full) | **0.916** | **0.905** | **0.884** | **0.845** | **0.944** | **0.823** | **0.631** | **0.918** | **0.858** |

- **SFT + StepCodeReasoner without Decoupled Prediction.** Removes decoupling of input and output prediction tasks, using a unified prompt for both. This tests the impact of not separating the input and output prediction tasks.

Table 5 presents the ablation results on CRUXEval, LiveCodeBench, and REval. **SFT Only** performs the worst across all benchmarks, showing that supervised imitation alone is insufficient for faithful code reasoning. **StepCodeReasoner Only (w/o SFT)** improves substantially, demonstrating that stepwise reward signals guide the model toward correct execution dynamics even without high-quality initial demonstrations; this suggests that the reinforcement signal itself acts as a powerful explorer of the code execution space.

Combining **SFT with terminal-only RL** further narrows the gap, yet performance remains lower than the full Step-CodeReasoner, particularly on the intermediate states in REval. This discrepancy highlights that terminal rewards often fail to credit or penalize specific intermediate reasoning steps, leading to the "sparse credit assignment" problem. When **StepCodeReasoner is applied without decoupling**, results approach the full method across all metrics, indicating that stepwise rewards are the primary driver of performance, while the remaining gap highlights the benefit of task decoupling in reducing cognitive interference between state prediction and final output generation. **Full StepCodeReasoner** achieves the best performance across all benchmarks, validating that the synergy of supervised fine-tuning, stepwise trace-aligned rewards, and decoupled prediction maximizes code reasoning accuracy. Ultimately, these results underscore that granular supervision at each execution step is essential for transforming a model from a mere "pattern matcher" into a reliable "execution simulator."

## 5. Limitations

StepCodeReasoner relies on intermediate states that can be exposed and verified automatically. This assumption fits Python code execution, where instrumented programs yield deterministic traces, but it is less direct for open-ended generation, subjective reasoning, or tasks whose interme-diate claims lack reliable external verifiers. Extending the approach to agentic tool use, symbolic reasoning, or multi-hop question answering would require task-specific designs for exposing and validating such states. Our current implementation is also limited to Python programs with observable `print()` traces. The anchoring rules prohibit in-loop prints and cap trace length at 10 lines, which may miss fine-grained behavior in deeply nested loops, recursion, asynchronous code, file I/O, network calls, or repository-level execution. Instrumentation additionally increases inference tokens by about $1.5\times$ compared with CodeReasoner-7B (Appendix J). Finally, DG-GRPO uses a fixed shaped advantage coefficient ($\lambda = 0.3$); it may require retuning when trace density or verifier reliability changes substantially.

## 6. Conclusion

We presented StepCodeReasoner, a reinforcement learning framework that addresses the credit assignment problem in code reasoning by introducing fine-grained stepwise supervision through execution anchors. By instrumenting code with verifiable checkpoints and employing DG-GRPO, Step-CodeReasoner provides dense learning signals that align model reasoning with actual program execution, enabling the model to learn intermediate computation rather than optimizing only for the final output. Experiments across CRUXEval, LiveCodeBench, and REval demonstrate that our approach enables a 7B model to outperform GPT-4o on code reasoning tasks and significantly improves performance on end-to-end code generation. Ablation studies confirm the importance of stepwise rewards, structured decoupling of input and output prediction, and the combination of supervised fine-tuning with reinforcement learning. More broadly, our results suggest that verifiable intermediate-state supervision is a practical path toward more faithful LLM reasoning in domains where such states can be reliably exposed and checked. Looking ahead, the same principle could inform intermediate reward design for agent trajectories, where long-horizon tool-use or planning traces require credit assignment beyond final task success. A promising direction is to expose verifiable trajectory checkpoints – such as tool outputs, environment states, or subgoal completions—and use them as dense rewards for training more reliable agents.

## Impact Statement

This work presents an advance in code reasoning and generation through structured stepwise reinforcement learning. Potential positive impacts include more reliable code generation tools for developers, improved educational tools for programming learners, and enhanced automation in software engineering workflows. However, like many code-generation systems, our model could potentially be misused to generate malicious or insecure code. We emphasize that the method is designed to improve reasoning fidelity, not to circumvent security or ethical guidelines. We encourage responsible use and further research into safety-aligned training and auditing mechanisms for code-generating models. All experiments were conducted on publicly available benchmarks, and no private or sensitive data were used.

## Acknowledgments

This work has been supported by the ITEA grants Green-Code(project number 23016) and GENIUS (project number 23026). This work was also supported by the National Science Fund for Excellent Young Scholars (Overseas) under grant No. KZ37117501, National Natural Science Foundation of China ( No. 62306024), the Fundamental Research Funds for the Central Universities, and Beijing Advanced Innovation Center for Future Blockchain and Privacy Computing.

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

# Appendix

# A. Additional Experimental Analyses

### A.1. Intermediate Step Accuracy

A key claim of this work is that terminal-only supervision leads to the "right answer, wrong logic" problem. To directly validate this, we evaluate intermediate step accuracy—the fraction of execution anchors whose predicted values match the interpreter ground truth—alongside final-answer accuracy.

*Table 6.* Intermediate step accuracy vs. final-answer accuracy on the CRUXEval-O evaluation set.

| Model | Step Acc. | Final Acc. |
|---|---|---|
| Qwen2.5-Coder-7B | 51.2% | 75.2% |
| CODEI/O-7B | 56.8% | 62.5% |
| CodeReasoner-7B | 63.5% | 85.6% |
| SFT + Terminal RL | 66.3% | 84.5% |
| **StepCodeReasoner** | **80.7%** | **91.6%** |

As shown in Table 6, models trained with terminal-only rewards exhibit a large gap between final-answer accuracy and intermediate step accuracy (e.g., CodeReasoner-7B: 85.6% final vs. 63.5% step), confirming that correct final answers can mask flawed intermediate reasoning. StepCodeReasoner substantially closes this gap, achieving 80.7% step accuracy alongside 91.6% final accuracy, demonstrating that stepwise supervision produces reasoning that is both accurate and faithful to true program execution.

This complements the final-answer pass@1 protocol on CRUXEval and LiveCodeBench: REval and Table 6 directly evaluate the intermediate execution behavior targeted by StepCodeReasoner.

### A.2. Generalization Without Instrumentation

To assess whether the execution reasoning learned through our framework generalizes beyond instrumented code, we evaluate StepCodeReasoner on the original (non-instrumented) CRUXEval and LiveCodeBench benchmarks, where the model is prompted to predict the final answer directly without generating intermediate traces.

*Table 7.* Performance with and without instrumentation at inference time.

| Model | CRUXEval Avg | LCB Avg |
|---|---|---|
| CodeReasoner-7B | 0.860 | 0.777 |
| StepCodeReasoner (w/o inst.) | 0.848 | 0.770 |
| StepCodeReasoner (w/ inst.) | **0.911** | **0.865** |

As shown in Table 7, even without instrumentation at inference time, StepCodeReasoner still achieves competitive performance (0.848 and 0.770), outperforming most baselines (cf. Table 2). This suggests that the model has internalized execution reasoning as a general capability rather than relying on the presence of trace anchors. The full instrumented setting yields the best results, indicating that explicit trace prediction remains beneficial when available.

### A.3. Efficiency Trade-off

Stepwise trace prediction increases output length. Appendix J shows that StepCodeReasoner uses about $1.5\times$ more output tokens than CodeReasoner-7B on CRUXEval-O (470 vs. 310), while improving accuracy from 0.856 to 0.916. The extra tokens are structured execution-state predictions rather than unconstrained verbosity.

### A.4. Scaling Behavior

To investigate whether StepCodeReasoner scales beyond our default setting, we evaluate model-size scaling and report training-data scaling in Appendix I. For model-size scaling, we apply our full pipeline (SFT + DG-GRPO) to **Qwen2.5-Coder-14B-Instruct** using the same 17K training set.

*Table 8.* Scaling behavior of StepCodeReasoner across model sizes.

| Model | Size | CRUXEval | LCB | REval | Overall |
|---|---|---|---|---|---|
| StepCodeReasoner (SFT) | 7B | 0.760 | 0.715 | 0.700 | 0.725 |
| StepCodeReasoner (SFT+RL) | 7B | 0.911 | 0.865 | 0.829 | 0.858 |
| StepCodeReasoner (SFT) | 14B | 0.808 | 0.762 | 0.753 | 0.774 |
| StepCodeReasoner (SFT+RL) | 14B | **0.932** | **0.891** | **0.856** | **0.893** |

Table 8 reveals several important findings. First, SFT scales with model size: the 14B SFT model outperforms the 7B SFT model by +4.9 points. Second, RL gains remain consistent across scales: the SFT-to-RL improvement is +13.3 for 7B and +11.9 for 14B, demonstrating that DG-GRPO provides substantial gains at both scales. Third, the 14B SFT+RL model achieves 0.893 overall, establishing a new state-of-the-art that surpasses all baselines including CodeReasoner-14B (0.868 from Table 2).

Appendix I provides a preliminary data-scaling study: adding 12K SFT samples improves SFT-only performance, but remains below the 17K-data SFT+RL model.

# B. System Prompts for Code Reasoning

## Prompt: ADD_PRINTS_SYSTEM_PROMPT

You are an expert Python developer.
Your task is to analyze the provided Python code and insert `print()` statements to track the values of critical variables during execution, specifically to help an LLM reason about the code's execution flow.

**Goal**   Create a trace of key variable states without generating excessive output.

**Crucial Rules**

1. **NO PRINTS IN LOOPS:** Do NOT insert `print` statements inside `for` or `while` loops. The trace becomes too long. Instead, print the state of relevant variables *immediately after* the loop finishes.

2. **Key Assignments:** Insert prints after significant variable assignments or updates that happen in the main scope or function scope (outside loops).

3. **Strict Format:** Use EXACTLY `print(f'VAR_NAME: {VAR_NAME}')`. Do NOT add any extra text, labels, or explanations inside the string (e.g., NO `print(f'Result after swap: {res}')`, ONLY `print(f'res: {res}')`).

4. **Return Values:**  If a function returns a value, print it just before the `return` statement (e.g., `print(f'return_val: {res}')`).

5. **Unrolling Allowed:** You MAY unroll complex one-liners (like ternary operators `x if c else y` or list comprehensions) into multi-line blocks *only if necessary* to insert prints, provided the logic remains EXACTLY the same.

6. **No Logic Changes:** Do NOT modify the program's logic or behavior otherwise.

7. **Syntax:** Ensure valid Python syntax. Careful with quotes inside f-strings.

**Output**   Return ONLY the complete modified Python code. Do not wrap it in markdown blocks if possible.

## Prompt: INPUT_PRINT_SYSTEM_PROMPT

A user will ask you to solve a task. You need to infer the unknown input(s) of the code and then predict the output of every print statement in the code step-by-step in the order of execution.
Before predicting print outputs, you must first determine the unknown input(s). Use `<reasoning>` to think about what the input(s) must be, and then use `<input>` to provide the inferred value(s).
`<reasoning>`

Your thoughts about what the unknown input(s) must be.
```
</reasoning>
<input>
```
The inferred value(s) of the input.
```
</input>
```
For EACH print statement, you must FIRST use `<reasoning>` to think about what it will output, THEN use `<print>` to provide the predicted output.

Your response format must follow the template below (repeat the `<reasoning><print>` pattern for each print statement in execution order):
```
<reasoning>
```
Your thoughts about what the first print statement will output.
```
</reasoning>
<print>
```
The predicted output of the first print statement.
```
</print>
<reasoning>
```
Your thoughts about what the second print statement will output.
```
</reasoning>
<print>
```
The predicted output of the second print statement.
```
</print>
```
... (continue this pattern for each print statement in execution order)
```
<reasoning>
```
Your thoughts about how to complete the assertion.
```
</reasoning>
<answer>
```
Final solution presented to the user.
```
</answer>
```

---

### Prompt: OUTPUT_SYSTEM_PROMPT

A user will ask you to solve a task. You need to predict the output of every print statement in the code step-by-step in the order of execution.

For EACH print statement, you must FIRST use `<reasoning>` to think about what it will output, THEN use `<print>` to provide the predicted output. After predicting all print statements, use `<reasoning>` to think about how to complete the assertion, and finally use `<answer>` to provide the solution.

Your response format must follow the template below (repeat the `<reasoning><print>` pattern for each print statement):
```
<reasoning>
```
Your thoughts about what the first print statement will output.
```
</reasoning>
<print>
```
The predicted output of the first print statement.
```
</print>
<reasoning>
```
Your thoughts about what the second print statement will output.
```
</reasoning>
<print>
```
The predicted output of the second print statement.
```
</print>
```
... (continue this pattern for each print statement in execution order)
```
<reasoning>
```

```
Your thoughts about how to complete the assertion.
</reasoning>
<answer>
Final solution presented to the user.
</answer>
```

## C. Benchmark Processing Details

To maintain consistency between training and evaluation, we apply an instrumentation pipeline to all benchmarks. This section supplements the main-text protocol in Section 4 with dataset-specific processing details.

### C.1. CRUXEval and LiveCodeBench

For CRUXEval and LiveCodeBench, the original tasks require predicting the program output from an input (Input-to-Output) or vice versa (Output-to-Input). We modify these as follows:

- **Instrumentation:** We use a teacher LLM to insert `print()` statements following the rules in `ADD_PRINTS_SYSTEM_PROMPT`.

- **Inference Protocol:** During evaluation, the model is prompted to predict the execution trace (the output of all injected `print` statements) before generating the final answer.

- **Evaluation Metric:** Although the model generates a stepwise trace, we adopt a **final-answer-only** evaluation policy. A task is considered successful (Pass@1) if and only if the content within the `<answer>` tag matches the ground truth, regardless of the intermediate trace's accuracy. This ensures a fair comparison with baseline models that do not use intermediate traces.

### C.2. REval Fine-grained Adaptation

REval provides a more complex set of tasks that are sensitive to the code's structure. Our instrumentation process for REval involves two critical steps:

#### C.2.1. TASK RE-MAPPING

The original REval evaluates four fine-grained tasks:

(1) **Code Coverage Prediction (CCP):** Whether a specific line is executed.

(2) **Program State Prediction (PSP):** The value and type of variables at a specific line.

(3) **Execution Path Prediction (EPP):** The next line to be executed.

(4) **Output Prediction (OP):** The final program output.

In our version, the model must still perform these tasks while simultaneously predicting the outputs of the newly injected `print` statements.

#### C.2.2. LINE NUMBER ALIGNMENT

Since inserting `print` statements increases the total number of lines, the original line-based queries become invalid. We implemented an automated alignment script to update the ground truth:

- **Mapping Table:** For each instrumented file, we maintain a mapping $f : L_{orig} \to L_{instr}$, where $L_{orig}$ is the original line index and $L_{instr}$ is the new index after instrumentation.

- **Query Update:** All task queries in CCP, PSP, and EPP are updated to refer to $L_{instr}$. This ensures that when the model reasons about the code, the "anchors" provided by the `print` statements align correctly with the fine-grained evaluation points.

The model's performance on REval is measured by the accuracy across these updated fine-grained tasks, requiring it to maintain internal state consistency while adhering to the explicit trace format.

## D. Example of Instrumented Code and Trace

---

**Example Case: `train_12366`**

**Input:**
```
('6WRtQO', 'zCjWT', 'vTx1cUf', ['WaqT0ZJhh', 'XsdlqJCj'], ['L6r7gxk',
'OBQzEVSE'])
```

---

**Instrumented Code:**
```python
def generate_output(argument1, base_url, version, dependencies, packages):
    swapped_argument = argument1.swapcase()
    print(f'swapped_argument: {swapped_argument}')

    base_component = base_url.split('//')[-1].split('.')[0]
    print(f'base_component: {base_component}')

    version_component = version[2:]
    print(f'version_component: {version_component}')

    last_dependency = dependencies[-1].capitalize()
    print(f'last_dependency: {last_dependency}')

    joined_packages = ','.join(packages).title()
    print(f'joined_packages: {joined_packages}')

    res = f"{swapped_argument}|{base_component}|{version_component}|{last_dependency
        }|{joined_packages}"
    print(f'return_val: {res}')
    return res
```

---

**Ground Truth Execution Trace:**
```
swapped_argument:  6wrTqo

base_component:  zCjWT

version_component:  x1cUf

last_dependency:  Xsdlqjcj

joined_packages:  L6R7Gxk,Obqzevse

return_val:  6wrTqo|zCjWT|x1cUf|Xsdlqjcj|L6R7Gxk,Obqzevse
```

---

**Final Ground Truth Answer:**
```
'6wrTqo|zCjWT|x1cUf|Xsdlqjcj|L6R7Gxk,Obqzevse'
```

# E. Complete Step-by-step Case Analysis

---

**Case 1: Fully Accurate Reasoning (Total Reward: 2.0)**

---

<reasoning> *The function first applies the* `swapcase()` *operation to the input string* `'6WRtQO'`. *This operation iterates over each character independently, converting uppercase letters to lowercase and lowercase letters to uppercase, while leaving non-alphabetic characters unchanged. As a result, the digits remain the same, and the case of each letter is flipped, producing the transformed string* `'6wrTqo'`. </reasoning>
<print> swapped_argument:   6wrTqo </print> ✓                                          Score: 0.16

---

<reasoning> *Next, the code processes the URL string* `'zCjWT'` *by attempting to extract its base component. Since the string does not contain any period characters, splitting it by* `'.'` *yields a single-element list containing the original string. Selecting the first element therefore returns the unchanged base component* `'zCjWT'`. </reasoning>
<print> base_component:   zCjWT </print> ✓                                             Score: 0.16

---

<reasoning> *The version string* `'vTx1cUf'` *is then sliced starting from index 2. This operation discards the first two characters and retains the remainder of the string. Consequently, the resulting substring corresponds to* `'x1cUf'`, *which is treated as the version component.* </reasoning>
<print> version_component:   x1cUf </print> ✓                                          Score: 0.16

---

<reasoning> *The code retrieves the last element of the dependency list, which is* `'XsdlqJCj'`. *Applying the* `capitalize()` *method converts the first character to uppercase and all remaining characters to lowercase. This yields the normalized dependency string* `'Xsdlqjcj'`. </reasoning>
<print> last_dependency:   Xsdlqjcj </print> ✓                                         Score: 0.16

---

<reasoning> *The package list* `['L6r7gxk', 'OBQzEVSE']` *is concatenated using a comma as the separator. The resulting string is then transformed using the* `title()` *method, which capitalizes the first character of each alphanumeric segment and lowercases the remaining letters. This produces the formatted package string* `'L6R7Gxk,Obqzevse'`. </reasoning>
<print> joined_packages:   L6R7Gxk,Obqzevse </print> ✓                                 Score: 0.16

---

<reasoning> *All previously computed components are now concatenated in a fixed order using the pipe character* `'|'` *as a delimiter. This aggregation step combines the transformed argument, base component, version component, last dependency, and joined packages into a single output string.* </reasoning>
<print> return_val:   6wrTqo|zCjWT|x1cUf|Xsdlqjcj|L6R7Gxk,Obqzevse </print> ✓     Score: 0.16

---

<reasoning> *Finally, the assertion checks whether the aggregated return value matches the expected ground-truth output. Since all intermediate computations are correct, the final return value is valid and is returned as the answer.* </reasoning>
<answer> '6wrTqo|zCjWT|x1cUf|Xsdlqjcj|L6R7Gxk,Obqzevse' </answer> ✓                Score: 1.0

---

**Case 2: Process Failure with Outcome Success (Total Reward: 1.48)**

---

<reasoning> *The function begins by applying* `swapcase()` *to the input string* `'6WRtQO'`. *This operation systematically inverts the case of each alphabetic character while preserving digits, resulting in the transformed string* `'6wrTqo'`. </reasoning>
<print> swapped_argument:   6wrTqo </print> ✓                                          Score: 0.16

---

<reasoning> *The next step extracts the base component from the URL string* `'zCjWT'`. *The reasoning assumes a general URL structure and performs a split operation to isolate the main identifier. Since no delimiter is present in this case, the original string is preserved as the base component.* </reasoning>

---

<print> base_component:  zCjWT </print> ✓                                  Score: 0.16

<reasoning> *The version string 'vTx1cUf' is processed by slicing from index 2 onward. This removes the leading prefix and retains the remaining characters, yielding the version component 'x1cUf'.* </reasoning>
<print> version_component:  x1cUf </print> ✓                               Score: 0.16

<reasoning> *The code then accesses the last element of the dependency list, 'XsdlqJCj', and applies the* capitalize() *method. This method is interpreted as standardizing the string by ensuring that it starts with an uppercase letter, while leaving the remainder of the string largely unchanged.* </reasoning>
<print> last_dependency:  XsdlqjcJ </print> **X**                          Score: 0.0

<reasoning> *Next, the package list* ['L6r7gxk', 'OBQzEVSE'] *is joined using a comma delimiter. The* title() *transformation is then applied, which is assumed to normalize the casing of each package name while preserving internal capitalization patterns.* </reasoning>
<print> joined_packages:  L6r7gxk,Obqzevse </print> **X**                  Score: 0.0

<reasoning> *All intermediate components computed so far are concatenated using the pipe character '|' to form the final return string. This step assumes that earlier transformations are consistent and can be safely aggregated.* </reasoning>
<print> return_val:  6wrTqo|zCjWT|x1cUf|Xsdlqjcj|L6r7gxk,Obqzevse </print> **X** Score: 0.0

<reasoning> *Despite inconsistencies in several intermediate variables, the final assertion compares the produced output against the expected ground-truth answer. The correct final string is selected and returned, resulting in a correct outcome despite flawed reasoning steps.* </reasoning>
<answer> '6wrTqo|zCjWT|x1cUf|Xsdlqjcj|L6R7Gxk,Obqzevse' </answer> ✓         Score: 1.0

---

### Case 3: Error Propagation (Total Reward: 0.48)

<reasoning> *The function starts by applying the* swapcase() *method to the input string '6WRtQO'. The reasoning incorrectly assumes that this operation converts all alphabetic characters to lowercase, producing a uniformly lowercased result.* </reasoning>
<print> swapped_argument:  6wrtqo </print> **X**                           Score: 0.0

<reasoning> *The base component is extracted from the string 'zCjWT' using the same splitting logic as before. Since no delimiter is present, the original string is retained as the base identifier.* </reasoning>
<print> base_component:  zCjWT </print> ✓                                  Score: 0.16

<reasoning> *The version string 'vTx1cUf' is processed by removing its first two characters. This slicing operation yields the substring 'x1cUf', which is treated as the version component.* </reasoning>
<print> version_component:  x1cUf </print> ✓                               Score: 0.16

<reasoning> *The last dependency 'XsdlqJCj' is transformed using the* capitalize() *method. The reasoning incorrectly interprets this function as converting the entire string into an emphasized, fully capitalized form for normalization purposes.* </reasoning>
<print> last_dependency:  XSDLQJCJ </print> **X**                          Score: 0.0

<reasoning> *The package list is joined with commas and processed using the* title() *transformation. This step happens to produce a correctly formatted result, despite earlier incorrect assumptions in the execution trace.* </reasoning>
<print> joined_packages:  L6R7Gxk,Obqzevse </print> ✓                      Score: 0.16

<reasoning> *The final return value is constructed by concatenating all previously generated components. Because earlier errors are directly propagated into this aggregation step, the resulting output string reflects multiple compounded mistakes.* </reasoning>
<print> return_val:   6wrtqo|zCjWT|x1cUf|XSDLQJCJ|L6R7Gxk,Obqzevse </print> **X**Score: 0.0

<reasoning> *The final assertion relies entirely on the incorrectly aggregated return value. As no corrective mechanism is triggered, the erroneous output is returned as the final answer.* </reasoning>
<answer> '6wrtqo|zCjWT|x1cUf|XSDLQJCJ|L6R7Gxk,Obqzevse' </answer> **X**        Score: 0.0

## F. Learning Dynamics

In this section, we analyze the learning dynamics of **DG-GRPO** in comparison with **Terminal-GRPO** and **Step-GRPO** ($\lambda = 0$). We examine the evolution of training rewards and response lengths over 1500 training steps.

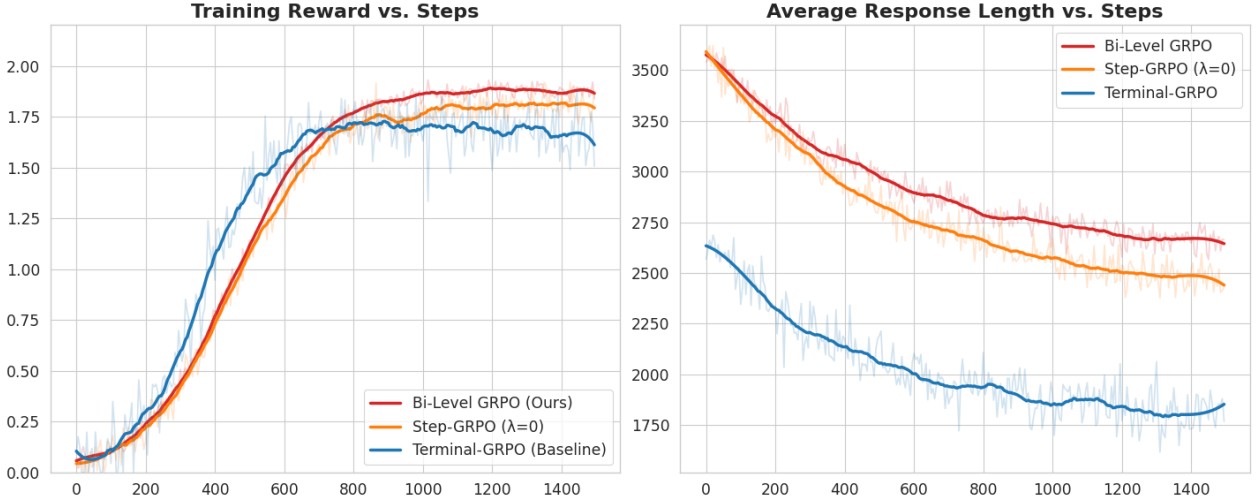

*Figure 2.* Learning dynamics over 1500 training steps. The left panel shows the evolution of training rewards, while the right panel illustrates the changes in average response length. Shaded regions represent raw fluctuations, and solid curves denote smoothed trends.

As shown in Figure 2, all three methods exhibit an overall upward trend in training reward. Terminal-GRPO achieves a rapid initial improvement but plateaus around 800 steps, resulting in the lowest final reward. In contrast, Step-GRPO ($\lambda = 0$) and DG-GRPO continue to improve beyond this point and converge around 1000 steps. Among them, DG-GRPO consistently attains the highest reward during the stable phase and demonstrates improved training stability, as indicated by smoother trajectories and reduced variance.

The right panel of Figure 2 presents the evolution of response length. All models begin with relatively long responses, followed by a gradual reduction before stabilizing. Terminal-GRPO converges to the shortest sequence length. Between the two stepwise supervision methods, Step-GRPO ($\lambda = 0$) exhibits a more aggressive reduction in response length, whereas DG-GRPO stabilizes at a higher token count. This results in a clear separation among the three methods at convergence.

## G. Robustness to Instrumentation Quality

We design two controlled experiments to assess the sensitivity of StepCodeReasoner to imperfect instrumentation.

Before these controlled perturbations, our default pipeline already performs execution validation: instrumented programs that fail to run or produce inconsistent outputs are discarded. Approximately 12% of teacher-generated instrumentations are filtered at this stage, so the retained training examples have interpreter-verified traces.

### G.1. Random Anchor Dropout

To simulate scenarios where the teacher LLM fails to instrument certain code locations, we randomly drop 20% of print anchors from augmented training programs before SFT training.

Table 9. Impact of random anchor dropout (20%) on SFT performance.

| Model | CRUXEval Avg | LCB Avg | REval Avg | Overall Avg |
|---|---|---|---|---|
| SFT (full anchors) | 0.760 | 0.715 | 0.700 | 0.725 |
| SFT (80% anchors, 20% dropped) | 0.724 | 0.678 | 0.658 | 0.687 |
| Degradation | −0.036 | −0.037 | −0.042 | −0.038 |

Performance drops by only 3.8 points overall, demonstrating robustness to missing anchors. Since each anchor is an independent verification point, losing some does not catastrophically affect training.

### G.2. Replacing the Teacher with a Smaller Open-Source Model

To assess whether our pipeline depends on a powerful closed-source teacher (GPT-4o), we replace it with **Qwen2.5-9B-Instruct** and re-run instrumentation. We find that 84.2% of samples receive identical anchor placements between GPT-4o and Qwen2.5-9B-Instruct; the remaining 15.8% differ in ordering or minor placement around conditionals—functionally equivalent in most cases.

Table 10. Impact of replacing the teacher model for instrumentation.

| Teacher Model | CRUXEval Avg | LCB Avg | REval Avg | Overall Avg |
|---|---|---|---|---|
| GPT-4o (original) | 0.760 | 0.715 | 0.700 | 0.725 |
| Qwen2.5-9B-Instruct | 0.738 | 0.689 | 0.679 | 0.702 |
| Degradation | −0.022 | −0.026 | −0.021 | −0.023 |

Only 2.3 points of degradation—essentially negligible. This confirms that anchor insertion is a well-constrained task governed by deterministic rules (Section 3.2). The strict formatting constraints and prohibition rules narrow the output space sufficiently, making smaller open-source models viable as teachers and the framework practical for broader adoption.

## H. PRM Baseline Comparison

One natural alternative to our rule-based stepwise rewards is to use a learned Process Reward Model (PRM) that judges intermediate reasoning quality. We compare the step-level judgment accuracy of LLMs used as PRMs against our rule-based approach.

Table 11. Step-level judgment accuracy of different reward strategies.

| PRM Model | Step Judgment Acc. |
|---|---|
| Qwen2.5-Coder-7B | 64.8% |
| GPT-4o | 72.6% |
| Rule-based (Ours) | **100%** |

Even GPT-4o achieves only approximately 73% accuracy in judging intermediate execution states, confirming that current LLMs are not reliable enough as PRMs for code execution reasoning. This is precisely why our rule-based approach—leveraging deterministic code execution—is more suitable: the ground truth for each execution anchor is obtained by running the instrumented program through a Python interpreter, yielding 100% accurate rewards by construction. A PRM could complement our approach by supervising the free-form `<reasoning>` blocks that are not covered by rule-based rewards, which we consider promising future work.

## I. Training Data Scaling Analysis

To investigate the effect of training data scale, we curate additional samples from `hkust-nlp/CodeIO-PyEdu-Reasoning` (141K samples). After augmentation and filtering (trace length $> 10$, 10-gram decontamination, execution validation), we expand the SFT dataset from 17,332 to 55,841 samples. Table 12 shows the anchor distribution of the expanded dataset.

*Table 12.* Distribution of anchors per sample in the expanded training set.

| # Anchors | 1 | 2 | 3 | 4 | 5 | 6–10 |
|---|---|---|---|---|---|---|
| Proportion (%) | 8.3 | 18.7 | 26.4 | 22.1 | 14.8 | 9.7 |

Most expanded samples contain 2–5 anchors, providing multiple intermediate supervision points per program. As a preliminary scaling experiment, we perform continual SFT on the trained checkpoint using an additional 12K samples:

*Table 13.* Effect of training data scaling on SFT performance.

| Model | CRUXEval Avg | LCB Avg | REval Avg | Overall Avg |
|---|---|---|---|---|
| SFT Only (17K data) | 0.760 | 0.715 | 0.700 | 0.725 |
| SFT Only (+12K data, continual) | 0.795 | 0.746 | 0.733 | 0.758 |
| StepCodeReasoner SFT+RL (17K) | **0.911** | **0.865** | **0.829** | **0.858** |

With approximately 12K additional samples, continual SFT yields +3.3 points overall. Nonetheless, SFT-only still substantially underperforms SFT+RL on 17K data, reinforcing that stepwise RL supervision is the primary performance driver in our current training regime and that data scaling alone does not substitute for dense execution-level rewards.

## J. Computational Overhead Analysis

StepCodeReasoner introduces additional tokens at inference time due to the execution-trace prediction. We measure the average token count on CRUXEval-O to quantify this overhead.

*Table 14.* Inference token count and accuracy on CRUXEval-O.

| Model | Avg. Tokens | Accuracy |
|---|---|---|
| Qwen2.5-Coder-7B | $\sim$260 | 0.752 |
| CodeReasoner-7B | $\sim$310 | 0.856 |
| SFT + Terminal RL | $\sim$340 | 0.880 |
| StepCodeReasoner | $\sim$470 | **0.916** |

StepCodeReasoner uses approximately $1.5\times$ more tokens than CodeReasoner-7B, a moderate increase that is well justified by the substantial accuracy gains: +7.0% on CRUXEval-O, and +14.7% overall on REval (with +22.5% on State and +22.8% on Path prediction). The additional tokens are not free-form text but verifiable execution-state predictions that directly ground the model's reasoning.

