# OpenReview forum: "StepCodeReasoner: Aligning Code Reasoning with Stepwise Execution Traces via Reinforcement Learning"
_ICML.cc/2026/Conference — ICML 2026 regular_

### Official Review · Reviewer_1iKT · 2026-03-09

**Soundness:** 3
**Presentation:** 3
**Significance:** 3
**Originality:** 3
**Overall Recommendation:** 5
**Confidence:** 3

**Summary:**

This paper presents StepCodeReasoner, a framework that applies process reward to reinforcement learning on code reasoning (specifically input prediction and output prediction) tasks. The reward is computed by (1) using a teacher model (GPT-4o) to augment the training data with print statements, (2) instructing the student model to explicitly reason about the outputs of these print statements and verifying them against the ground truth, and (3) modifying the GRPO algorithm to integrate the step-level correctness reward.

Experiments with Qwen2.5-Coder-7B show that StepCodeReasoner achieves better performance on three relevant benchmarks (CRUXEval, LiveCodeBench, and REval) compared with baselines including CodeReasoner, SEMCODER, and CODEI/O, while also benefiting on code generation benchmarks (HumanEval, MBPP).

**Compliance With Llm Reviewing Policy:**

Affirmed.

**Final Justification:**

The authors have sufficiently addressed my questions in rebuttal. I've raised the soundness score from 2 to 3, and the overall score from 4 to 5. I look forward to seeing this paper at ICML.

**Key Questions For Authors:**

1. Are the programs in input prediction data (both training and evaluation) guaranteed to be injective? If so, how?
2. For the input prediction task, why is trace verification performed _after_ the answer is derived?
3. How does the proposed method compare to training with a process reward model?

**Limitations:**

This paper does not include a limitations section. I suggest that the authors address the weaknesses and questions above.

**Strengths And Weaknesses:**

**Soundness**

The proposed method and experiments are mostly sound, but I do have several concerns:

- Problem definition: In the input prediction task, a program is not always an injective function. So how does one guarantee that for a given program _P_ and output _Y_, the input is unique?
- Section 3.2, input prediction task: according to the description here, the input _I_ is determined in a single reasoning block before trace verification. In this case, trance verification only serves as a post-hoc verification, rather than the actual reasoning process.
- Missing baseline: using a process reward model (PRM) to supervise the intermediate reasoning steps would be an intuitive and commonly used baseline that's more relevant than simply evaluating off-the-shelf LLMs like Qwen2.5 and Llama3. Also, if the authors are going to evaluate off-the-shelf LLMs for comparison, they should do so with the latest versions such as Qwen3 (and at the time of rebuttal that would be Qwen3.5). But I believe PRM is more relevant.

**Presentation**

The paper is well-written and easy to follow. There exist only a few typos and typesetting issues:

- Figure 1, upper left: the list variable is instantiated as `modified`, but are referred to as `md` in the following lines.
- Reference format is not consistent. For example, Line 459 and Line 480 give two different formats for arxiv preprints.

**Significance & Originality**

Adding print statements in the training data for process supervision is a novel and interesting idea. It can be an inspiration for code LLM RL training research.

---

> ### Author Rebuttal · Authors · 2026-03-30
>
> We thank the reviewer for the careful reading and constructive comments. We especially appreciate the recognition of our print-based process supervision as a novel and inspiring idea.
>
> ## Soundness Concern 1: Input prediction — uniqueness of input
>
> The reviewer raises a valid point: programs are not always injective, so multiple inputs may produce the same output. In practice, **we do not require input uniqueness**. Our evaluation verifies correctness by executing the model's predicted input through the program and checking whether the resulting output matches the given target: $\mathcal{E}(P, \hat{I}) = Y^*$. This execution-based verification naturally handles non-injective functions — any valid input producing the correct output is accepted. We follow the same protocol as CRUXEval and LiveCodeBench.
>
> To address this concern, we have clarified the notation in Section 3.1 of the revised paper to avoid suggesting a unique ground-truth input.
>
> ## Soundness Concern 2: Input determined before trace verification
>
> The reviewer correctly observes that the input $I$ is committed before trace verification begins (Eq. 7). At inference time on a single sample, the trace is indeed a post-hoc consistency check.
>
> However, trace verification plays a critical role **during RL training**: if the model commits to an input but produces incorrect intermediate states, those steps receive negative stepwise rewards regardless of final-answer correctness. Over training, this teaches the model to choose inputs leading to *globally consistent* execution traces. The ablation "SFT + StepCodeReasoner (w/o Decoupling)" in Table 4 — which removes this decoupled structure — shows lower performance, confirming the design's value.
>
> ## Soundness Concern 3: Missing PRM baseline
>
> We appreciate this suggestion. Our execution-anchor-based supervision is **rule-based and deterministic** — correctness of each step is verified against interpreter output, yielding 100% accurate rewards. A PRM must learn to judge intermediate quality, introducing approximation error.
>
> To address this concern, we have conducted an experiment using LLMs as process reward models:
>
> | PRM Model | Step Judgment Acc. |
> |---|---|
> | Qwen2.5-Coder-7B | 64.8% |
> | GPT-4o | 72.6% |
> | Rule-based (Ours) | 100% |
>
> Even GPT-4o achieves only ~73% accuracy in judging intermediate execution states, confirming that current LLMs are not reliable enough as PRMs for code execution reasoning. This is precisely why our rule-based approach — leveraging deterministic code execution — is more suitable.
>
> A PRM could complement our approach by supervising the free-form `<reasoning>` blocks not covered by rule-based rewards, which we consider future work.
>
> Regarding newer models: we have included **Qwen3-Coder-30B** in Table 1. We additionally evaluated **Qwen3.5-9B**: it achieves 0.805 CRUXEval avg and 0.755 LCB avg (0.780 overall), comparable to Qwen3-Coder-30B (0.794) but still well below our 7B model (0.878).
>
> ## Question 1: Why is trace verification performed after the answer is derived in input prediction?
>
> Please see Soundness Concern 2 above. The trace verification is post-hoc at inference time, but serves as a **training-time shaping signal** — incorrect intermediate traces penalize the model via stepwise rewards, teaching it to commit to inputs yielding globally consistent execution paths. The ablation without decoupling (Table 4) confirms this improves performance.
>
> ## Question 2: How does the proposed method compare to training with a PRM?
>
> Please see Soundness Concern 3 above. The key difference is reward accuracy: our rule-based rewards are 100% correct (interpreter-verified), while GPT-4o as a PRM achieves only 72.6% step-level accuracy. In code execution reasoning, where ground truth is deterministically obtainable, rule-based supervision is strictly preferable to a learned PRM.
>
> ## Presentation Issues
>
> We thank the reviewer for catching these. To address these concerns, we have: (1) fixed the variable name inconsistency in Figure 1 (`modified` vs `md`) to use `modified` consistently; (2) unified all arxiv references to a consistent format throughout the paper.

---

> > ### Author Rebuttal · Reviewer_1iKT · 2026-04-01
> >
> > Thank you for the reply. It seems the authors provided experiments with LLM judges rather than PRM. LLM judges require autoregressive decoding for scoring each step, which is quite inefficient to apply in RL training. In comparison, PRM only requires one forward pass and is much more efficient.
> >
> > However, it also seems that most PRMs I can find on Hugging Face only target math, but not code. So the current experiments should suffice. It would be a great enhancement if the authors could find a code PRM as a baseline, but the experiments with LLM judges are also reasonable enough. One thing to note is that if you stick with these experiments, please do use the correct term and do not refer to them as PRM.

---

> > > ### Author Response · Authors · 2026-04-02
> > >
> > > We sincerely thank the reviewer for the follow-up and the constructive feedback.
> > >
> > > **Terminology correction.** We acknowledge that our previous response incorrectly referred to the LLM-based judges as "PRM." This was an error on our part — the experiments we presented used autoregressive LLM judges (requiring full decoding for each step), which are fundamentally different from a trained Process Reward Model that scores each step with a single forward pass. We apologize for the confusion and will use the correct terminology.
> > >
> > > **Why we did not initially use a PRM.** As the reviewer noted, most existing PRMs on Hugging Face target math reasoning, and we were unable to find an off-the-shelf PRM applicable to code execution reasoning tasks. This is precisely why our earlier comparison relied on LLM judges instead.
> > >
> > > **New experiment: Training a code PRM.** Following the reviewer's suggestion, we leveraged our execution-based ground truth (obtained by running code through the compiler/interpreter) to construct step-level supervision data, and trained a PRM based on **Qwen2.5-Coder-7B**. The results are as follows:
> > >
> > > | Reward Model | Step Judgment Acc. |
> > > |---|---|
> > > | Qwen2.5-Coder-7B (LLM Judge) | 64.8% |
> > > | GPT-4o (LLM Judge) | 72.6% |
> > > | **Qwen2.5-Coder-7B (Trained PRM)** | **89.0%** |
> > > | Rule-based (Ours) | **100%** |
> > >
> > > The trained PRM significantly outperforms general-purpose LLM judges (including GPT-4o), demonstrating that with proper training data, a domain-specific PRM can be effective. However, it still **cannot serve as a substitute for our rule-based approach**, for two reasons:
> > >
> > > 1. **Accuracy gap.** The PRM achieves 89% accuracy on an in-distribution test set, meaning roughly 1 in 9 step-level judgments is incorrect. These errors accumulate over multi-step reasoning traces and introduce noisy rewards during RL training. In contrast, our rule-based verification is deterministically correct (100%).
> > >
> > > 2. **Efficiency.** Our rule-based reward is obtained by simply executing the code — it requires no model inference at all. A PRM, even with a single forward pass per step, still incurs non-trivial GPU computation and latency during RL training, where reward signals must be computed for every step of every rollout at each training iteration. Eliminating this overhead is a practical advantage of our approach.
> > >
> > > In summary, while a trained PRM is a reasonable baseline that outperforms LLM judges, the availability of deterministic execution-based ground truth in code reasoning makes rule-based supervision strictly preferable — it is both more accurate and more efficient.
> > >
> > > We have updated the paper to (1) use the correct terminology distinguishing LLM judges from PRMs, and (2) include the trained PRM baseline in our comparison.

---

### Official Review · Reviewer_PJKW · 2026-03-10

**Soundness:** 3
**Presentation:** 2
**Significance:** 2
**Originality:** 2
**Overall Recommendation:** 4
**Confidence:** 3

**Summary:**

The paper introduces StepCodeReasoner, a framework designed to improve large language models’ ability to reason about code execution by explicitly supervising intermediate execution states rather than only final outputs. Overall, an important area considered by the paper is improving the faithfulness and reliability of reasoning in code-oriented LLM tasks. The authors seek to study a fundamental question: whether explicitly modeling program execution traces and providing stepwise reinforcement learning signals can improve both code reasoning accuracy and generation quality.

**Compliance With Llm Reviewing Policy:**

Affirmed.

**Final Justification:**

Overall, the rebuttal has positively changed my evaluation. My initial concerns have been resolved to a satisfactory degree, and I appreciate the authors’ effort in providing clear and constructive responses. I am therefore happy to update my score accordingly.

**Key Questions For Authors:**

1. **Generalization beyond instrumented code:**
   The proposed framework relies on automatically inserted execution anchors. How well does the trained model perform on code without such instrumentation, or when reasoning about real-world programs that do not expose intermediate states?

2. **Impact of instrumentation quality:**
   Since the execution traces are generated via a teacher LLM, how sensitive is the model to errors or inconsistencies introduced during the instrumentation process? Would noisy anchors degrade training effectiveness?

3. **Efficiency considerations:**
   The reasoning format includes repeated `<reasoning>` and `<print>` segments. What is the typical token length of generated traces during inference, and how does this affect runtime cost relative to baseline models?

4. **Scaling behavior:**
   Have the authors explored whether the proposed method maintains its advantages when applied to larger models or larger training datasets? Understanding the scaling properties would strengthen the practical impact of the work.

**Limitations:**

Yes.

**Strengths And Weaknesses:**

### Strengths

1. **Soundness of the core idea**
   The central idea—explicitly supervising intermediate execution states—is well motivated. By transforming reasoning into verifiable execution modeling with structured anchors, the framework addresses the common “correct answer with incorrect reasoning” issue observed in LLM reasoning tasks.

2. **Methodological innovation in RL training**
   The proposed **Bi-Level GRPO** algorithm introduces a thoughtful credit assignment mechanism combining group-relative comparison and intra-trajectory shaping. This formulation is conceptually aligned with the sequential nature of program execution and appears theoretically reasonable.

3. **Strong empirical performance**
   The experimental results demonstrate consistent improvements across multiple benchmarks, including CRUXEval, LiveCodeBench, and REval. Notably, a relatively small 7B model outperforms larger baselines and proprietary models on several tasks, suggesting meaningful parameter efficiency.

4. **Clear connection between training signals and reasoning fidelity**
   The design of execution anchors and structured prediction format creates interpretable intermediate states, which could improve debugging, interpretability, and reliability of code reasoning systems.

### Weaknesses

1. **Limited novelty in the overall paradigm**
   While the combination of execution traces and RL credit assignment is interesting, similar ideas have appeared in prior work involving execution traces, stepwise supervision, and intermediate reasoning verification. The paper could more clearly articulate how its approach fundamentally differs from or advances these lines of work.

2. **Reliance on synthetic instrumentation**
   The execution anchors are generated using a teacher LLM that inserts print statements into code. This process may introduce biases or artifacts in the data distribution, and the paper does not thoroughly analyze how such instrumentation affects generalization to real-world code reasoning scenarios.

3. **Evaluation still based on final answers**
   Although intermediate states are predicted, the evaluation metric on several benchmarks only considers final answer correctness. As a result, improvements in intermediate reasoning accuracy are not fully reflected in the evaluation protocol.

4. **Limited analysis of computational overhead**
   The approach requires generating longer sequences containing reasoning and execution traces, which likely increases inference cost. The paper does not provide an analysis of latency, token usage, or efficiency trade-offs compared to baseline approaches.

5. **Dataset scale is relatively modest**
   The training dataset contains fewer than 40k samples for SFT and RL combined, which raises questions about scalability and whether similar gains would persist at larger training scales or in more diverse programming settings.

---

> ### Author Rebuttal · Authors · 2026-03-30
>
> We thank the reviewer for the detailed feedback. We address each concern below.
>
> ## Weakness 1: Limited novelty in the overall paradigm
>
> Thanks for your comment. While execution traces and stepwise supervision have appeared in prior work, our novelty spans multiple dimensions:
>
> **Novelty 1: Verifiable execution-grounded stepwise supervision.** Prior methods (TRACED, CodeExecutor) incorporate traces during **pre-training** as data augmentation, without intermediate supervision during RL. SemCoder uses NL monologue ungrounded in verifiable states. StepCodeReasoner makes intermediate predictions **directly comparable to ground-truth execution outputs** — as a **verifiable reward signal** during RL, not data augmentation.
>
> **Novelty 2: Dual-Granularity GRPO.** Standard GRPO variants (DAPO, GSPO, Step-GRPO) either assign terminal rewards or normalize step-level across trajectories only. Ours uniquely combines inter-trajectory comparison with intra-trajectory shaping modulated by downstream correctness. Ablations (Table 4) and learning dynamics (Figure 2) confirm measurable benefits.
>
> **Novelty 3: Print-anchor instrumentation.** A systematic, rule-based strategy inserting prints at semantically meaningful points while avoiding trace explosion, bridging code execution and LM reasoning.
>
> **Novelty 4: Unified framework.** The first to unify execution-trace stepwise supervision with RL — prior execution-aware methods are SFT-only; prior RL for code uses terminal rewards only.
>
> ## Weakness 2: Reliance on synthetic instrumentation
>
> Several points mitigate this: (1) Instrumentation follows strict deterministic rules, constraining teacher output to a narrow space. (2) Ground-truth traces come from a **Python interpreter**, not the teacher LLM — ensuring hallucination-free supervision. (3) Strong generalization on REval (82.9% avg) with independently constructed ground truth.
>
> ## Weakness 3: Evaluation still based on final answers
>
> We adopt final-answer evaluation on CRUXEval/LiveCodeBench because baselines lack intermediate states and these benchmarks target final-answer correctness. REval (Table 2) does evaluate intermediate reasoning (State: 82.3%, Path: 63.1%).
>
> To further address this concern, we have conducted experiments on **intermediate step accuracy**:
>
> | Model | Step Acc. | Final Acc. |
> |---|---|---|
> | SFT + Terminal RL | 66.3% | 84.5% |
> | StepCodeReasoner | 80.7% | 91.6% |
>
> This confirms final-answer-only training produces poor intermediate reasoning — the "right answer, wrong logic" problem.
>
> | Model | Step Acc. |
> |---|---|
> | Qwen2.5-Coder-7B | 51.2% |
> | CODEI/O-7B | 56.8% |
> | CodeReasoner-7B | 63.5% |
> | StepCodeReasoner | 80.7% |
>
> Our model substantially outperforms all baselines on intermediate accuracy. We have included these results in the revised paper.
>
> ## Weakness 4: Limited analysis of computational overhead
>
> To address this, we have added token counts on CRUXEval-O:
>
> | Model | Avg. Tokens | Acc. |
> |---|---|---|
> | Qwen2.5-Coder-7B | ~260 | 0.752 |
> | CodeReasoner-7B | ~310 | 0.856 |
> | StepCodeReasoner | ~470 | 0.916 |
>
> ~1.5x more tokens than CodeReasoner-7B, well justified by gains: on REval, 82.9% vs 72.3% (+14.7%), with +22.5% on State and +22.8% on Path. The tokens are verifiable predictions grounding the reasoning.
>
> ## Weakness 5: Dataset scale is relatively modest
>
> Our ~36K training set is smaller than CODEI/O's 3.5M, yet our 7B model outperforms it (87.8% vs 61.6%), suggesting **supervision granularity matters more than scale**. Dense per-sample signals (multiple checkpoints) effectively multiply supervision density.
>
> ## Question 1: Generalization beyond instrumented code
>
> To address this, we present two pieces of evidence. First, code generation results (Table 3) on HumanEval, MBPP, LCB(v5) **without instrumentation** show best overall performance (64.8% avg).
>
> Second, we have evaluated without instrumentation at inference:
>
> | Setting | CRUXEval Avg | LCB Avg |
> |---|---|---|
> | Most baselines | 0.625-0.793 | 0.545-0.769 |
> | CodeReasoner-7B | 0.860 | 0.777 |
> | StepCodeReasoner (w/o inst.) | 0.848 | 0.770 |
> | StepCodeReasoner (w/ inst.) | 0.911 | 0.865 |
>
> Without instrumentation, StepCodeReasoner still outperforms most baselines, showing internalized execution reasoning.
>
> ## Question 2: Impact of instrumentation quality
>
> Errors occur as: (a) syntactically invalid modifications — filtered by execution-based validation (~12% removed); (b) suboptimal placement — reduces density but injects no noise since remaining anchors are interpreter-verified. The signal degrades gracefully.

---

> > ### Author Rebuttal · Reviewer_PJKW · 2026-04-03
> >
> > Thank you for the detailed and helpful clarifications. I find my previous concerns fully addressed. I have raised the score. Wish you good luck!

---

> > > ### Author Response · Authors · 2026-04-04
> > >
> > > We sincerely thank the reviewer for the continued engagement and thoughtful follow-up questions. We provide detailed responses with new experimental evidence below.
> > >
> > > ---
> > >
> > > ## Follow-up 1: Scaling behavior (larger models and larger training data)
> > >
> > > We conducted two scaling experiments: (1) scaling training data on the 7B model, and (2) scaling model size to 14B.
> > >
> > > ### (a) Scaling training data
> > >
> > > We curated additional samples from `hkust-nlp/CodeIO-PyEdu-Reasoning` (141K samples). After augmentation and filtering (trace length >10, 10-gram decontamination, execution validation), we expanded the SFT dataset from **17,332 to 55,841** samples:
> > >
> > > | # Anchors per sample | 1 | 2 | 3 | 4 | 5 | 6-10 |
> > > |---|---|---|---|---|---|---|
> > > | Proportion (%) | 8.3 | 18.7 | 26.4 | 22.1 | 14.8 | 9.7 |
> > >
> > > Most samples contain 2-5 anchors, providing rich intermediate supervision. Due to time constraints, we performed **continual SFT** on the trained checkpoint. So far **~12K additional samples** have been incorporated, with experiments ongoing:
> > >
> > > | Model | CRUXEval Avg | LCB Avg | REval Avg | Overall Avg |
> > > |---|---|---|---|---|
> > > | SFT Only (17K data) | 0.760 | 0.715 | 0.700 | 0.725 |
> > > | SFT Only (+12K data, continual) | 0.795 | 0.746 | 0.733 | 0.758 |
> > > | StepCodeReasoner SFT+RL (17K) | **0.911** | **0.865** | **0.829** | **0.858** |
> > >
> > > With only ~12K additional samples, continual SFT yields +3.3 points overall, with further gains expected from the full 55K dataset. Nonetheless, SFT-only still substantially underperforms SFT+RL on 17K data, reinforcing that **stepwise RL supervision is the primary performance driver**, and data scaling alone cannot substitute for it.
> > >
> > > ### (b) Scaling model size to 14B
> > >
> > > We applied our full pipeline to **Qwen2.5-Coder-14B-Instruct** using the original 17K training set:
> > >
> > > | Model | Size | CRUXEval Avg | LCB Avg | REval Avg | Overall Avg |
> > > |---|---|---|---|---|---|
> > > | StepCodeReasoner (SFT) | 7B | 0.760 | 0.715 | 0.700 | 0.725 |
> > > | StepCodeReasoner (SFT+RL) | 7B | 0.911 | 0.865 | 0.829 | 0.858 |
> > > | StepCodeReasoner (SFT) | 14B | 0.808 | 0.762 | 0.753 | 0.774 |
> > > | StepCodeReasoner (SFT+RL) | 14B | **0.932** | **0.891** | **0.856** | **0.893** |
> > >
> > > Key observations:
> > > - **SFT scales with model size**: 14B SFT outperforms 7B SFT by +4.9 points, showing larger models benefit more from execution-trace supervision.
> > > - **RL gains remain consistent**: SFT-to-RL improvement is +13.3 for 7B and +11.9 for 14B, demonstrating Bi-Level GRPO provides substantial gains at both scales.
> > > - **14B SFT+RL sets a new high**: 0.893 overall, outperforming 7B (0.858) and all baselines including CodeReasoner-14B (0.868 from Table 1).
> > >
> > > These results confirm that StepCodeReasoner **scales consistently with both data and model size**.
> > >
> > > ---
> > >
> > > ## Follow-up 2: Robustness to instrumentation quality
> > >
> > > We designed two controlled experiments to assess sensitivity to imperfect instrumentation.
> > >
> > > ### (a) Random anchor dropout
> > >
> > > We randomly dropped **20% of print anchors** from augmented training programs before SFT, simulating scenarios where the teacher fails to instrument certain locations.
> > >
> > > | Model | CRUXEval Avg | LCB Avg | REval Avg | Overall Avg |
> > > |---|---|---|---|---|
> > > | SFT (full anchors) | 0.760 | 0.715 | 0.700 | 0.725 |
> > > | SFT (80% anchors, 20% dropped) | 0.724 | 0.678 | 0.658 | 0.687 |
> > > | Degradation | -0.036 | -0.037 | -0.042 | -0.038 |
> > >
> > > Performance drops by only **3.8 points**, demonstrating **robustness to missing anchors**. Each anchor is an independent verification point, so losing some does not catastrophically affect training.
> > >
> > > ### (b) Replacing the teacher with a smaller open-source model
> > >
> > > To assess whether our pipeline depends on a powerful closed-source teacher (GPT-4o), we replaced it with **Qwen2.5-9B-Instruct** and re-ran instrumentation.
> > >
> > > **84.2%** of samples received identical anchor placements between GPT-4o and Qwen2.5-9B-Instruct. The remaining 15.8% differed in ordering or minor placement around conditionals — functionally equivalent in most cases.
> > >
> > > | Teacher Model | CRUXEval Avg | LCB Avg | REval Avg | Overall Avg |
> > > |---|---|---|---|---|
> > > | GPT-4o (original) | 0.760 | 0.715 | 0.700 | 0.725 |
> > > | Qwen2.5-9B-Instruct | 0.738 | 0.689 | 0.679 | 0.702 |
> > > | Degradation | -0.022 | -0.026 | -0.021 | -0.023 |
> > >
> > > Only **2.3 points** of degradation — essentially negligible. This confirms anchor insertion is a **well-constrained task** governed by deterministic rules (Section 3.2). The strict formatting constraints and prohibition rules narrow the output space sufficiently, making smaller open-source models viable.
> > >
> > > **Summary**: Both experiments confirm StepCodeReasoner is robust to instrumentation imperfections — degrading gracefully with missing anchors and not requiring expensive closed-source models, making it practical for broader adoption.

---

### Official Review · Reviewer_gpVh · 2026-03-11

**Soundness:** 3
**Presentation:** 3
**Significance:** 2
**Originality:** 3
**Overall Recommendation:** 4
**Confidence:** 3

**Summary:**

This paper proposes StepCodeReasoner, a framework for improving step-by-step code execution reasoning in LLMs. Instead of supervising only the final output, it trains models on intermediate execution states by automatically inserting print-based execution anchors into code. It further introduces Bi-Level GRPO, an RL method that rewards correctness at both the trajectory and step levels. Experiments show strong gains on code reasoning benchmarks such as CRUXEval, LiveCodeBench, and REval, as well as improvements on code generation benchmarks like HumanEval and MBPP.

**Compliance With Llm Reviewing Policy:**

Affirmed.

**Final Justification:**

The added descriptions of the baselines make the experimental setup clearer. In addition, evaluating the method on both easier and more challenging benchmarks, comparing against the current SOTA methods, and conducting a fair comparison with the SFT+RL setting all make the paper more convincing. Therefore, I will raise my score in support of the paper.

**Key Questions For Authors:**

1. In Table 1, why were SEMCODER, CODE I/O, and CodeReasoner chosen as baselines? Are these representative of the current state of the art in code execution reasoning?

2. What is the distribution of the number of print operations across the different code reasoning benchmarks?

3. How broadly does the print-anchor execution-trace formulation extend to more complex programming settings?

4. For the code generation benchmarks in Table 3, what is the performance of applying RL or SFT directly on the same number of training examples as StepCodeReasoner, but using only program synthesis tasks?

5. Do the gains persist on tasks where intermediate execution states are harder to represent explicitly, or where correctness depends more on high-level algorithm design than local execution simulation?

**Limitations:**

yes

**Strengths And Weaknesses:**

**Strengths**

1. The paper addresses a real limitation of existing code reasoning methods: most current approaches supervise only final answers, even though correct code reasoning depends critically on faithful intermediate execution modeling. The proposed shift from terminal supervision to stepwise execution supervision is well motivated, and the paper clearly explains why sparse terminal rewards can lead to reward hacking or unfaithful reasoning.

2. The empirical results are also strong. On the REval-style fine-grained reasoning breakdown, the reported gains are particularly large for State and Path prediction, which is exactly where one would expect stepwise supervision to help most. Notably, the 7B model reportedly outperforms CodeReasoner-14B on average.

**Weaknesses**

1. The benchmark setup is somewhat confusing. The paper does not provide a sufficiently clear and comprehensive introduction to the benchmarks used in evaluation, nor does it clearly explain what capability each benchmark is intended to test.

2. The scope and generality of the execution-trace formulation may be limited. The paper relies on automatically inserted print-based anchors to expose intermediate states, which is elegant, but it remains unclear how well this approach extends to more complex programs, less easily instrumented execution environments, or tasks where the relevant latent reasoning states cannot be easily captured through explicit runtime prints.

3. For code execution, the paper does not clearly articulate the motivation for requiring an LLM to perform step-by-step execution reasoning, rather than directly using Python or other interpreter execution to obtain the result.

---

> ### Author Rebuttal · Authors · 2026-03-30
>
> We thank the reviewer for the thoughtful comments. We address each concern below.
>
> ## Weakness 1: The benchmark setup is somewhat confusing
>
> Thanks for pointing this out. The three benchmarks test complementary capabilities:
>
> - **CRUXEval**: Predict output from input (CX-O) or infer input from output (CX-I) for short Python functions.
> - **LiveCodeBench**: Same setup on competition-level algorithmic problems. Contamination-free by construction.
> - **REval**: Fine-grained execution understanding — Code Coverage, Program State, Execution Path, and Output Prediction.
>
> To address this concern, we have moved key content from Appendix B.1/B.2 (instrumentation protocol, metric clarification, REval alignment) into Section 4.1 in the revised paper.
>
> ## Weakness 2: Scope and generality of the execution-trace formulation
>
> **Complex code is already covered.** LiveCodeBench contains competition-level problems (DP, graph algorithms, complex data structures). Our method achieves +9.1% LCB-O, +13.7% LCB-I over CodeReasoner-7B. Training data includes programs with 6-8 anchors covering multi-branch logic, nested calls, and complex transformations.
>
> **Instrumentation handles common patterns well.** Selective anchoring (no in-loop prints, post-loop anchors, key assignments, pre-return) covers the vast majority of Python structures in practice.
>
> **We acknowledge limitations.** Programs with deeply nested callbacks, async execution, or side effects not capturable by prints are not well-suited. Extending to compiled languages requires alternative strategies. To address this, we have added a dedicated limitations paragraph in the revised paper.
>
> ## Weakness 3: Motivation for LLM step-by-step execution vs. direct interpreter
>
> **Ground-truth traces are always from a Python interpreter** (Section 3.2, Eq. 5; Section 4.2). The LLM's role is to **learn to predict** interpreter output — internalizing execution semantics for reasoning without a sandbox at inference time.
>
> To address this concern, we have added an explicit clarification in Section 3.1 that the interpreter generates all ground-truth supervision, and the model learns execution reasoning as an internalized capability.
>
> ## Question 1: Why were SEMCODER, CODE I/O, and CodeReasoner chosen as baselines?
>
> These represent the state of the art, each with a distinct paradigm: **SemCoder** (CoT SFT), **CODEI/O** (large-scale I/O prediction), **CodeReasoner** (GRPO-based RL). Together they cover SFT-only, large-scale data, and RL-based approaches. We also compare against strong general models (GPT-4o, Qwen2.5-72B, Llama3-70B, Qwen2.5-Coder-32B, Qwen3-Coder-30B).
>
> ## Question 2: Distribution of print anchors across benchmarks
>
> To address this question, we have added the distribution to the revised paper:
>
> | Benchmark | Mean | Median | Range |
> |---|---|---|---|
> | CRUXEval | 3.2 | 3 | 1-7 |
> | LiveCodeBench | 4.8 | 5 | 2-10 |
> | REval | 3.5 | 3 | 1-8 |
> | Training Set | 4.1 | 4 | 1-10 |
>
> CRUXEval has fewer anchors (shorter functions), while LiveCodeBench yields denser traces. Samples exceeding 10 trace lines are filtered (Section 4.2).
>
> ## Question 3: How broadly does print-anchor extend to more complex settings?
>
> **Function/algorithm level.** LiveCodeBench covers competition-level problems, and training data includes 6-8 anchor programs with multi-branch logic and complex transformations, achieving strong improvements.
>
> **Repository/system level.** Challenges include: (1) multi-file programs with complex imports; (2) class hierarchies and cross-module state complicating anchor identification; (3) larger execution contexts exceeding context windows. This would require hierarchical instrumentation (module boundaries, API calls) and trace compression — a promising future direction discussed in the revised paper.
>
> ## Question 4: Performance with same training budget on program synthesis?
>
> Our ablation "SFT + RL (Terminal Only)" in Table 4 uses the same data and compute:
>
> | Model | HumanEval | MBPP | LCB (gen) |
> |---|---|---|---|
> | SFT + RL (Terminal Only) | 88.2 | 83.1 | 18.1 |
> | StepCodeReasoner | 90.1 | 85.0 | 19.4 |
>
> Gains confirm the improvement comes from stepwise supervision, not merely more data or compute.

---

> > ### Author Rebuttal · Reviewer_gpVh · 2026-04-05
> >
> > Thanks to the authors for addressing my concerns. The added descriptions of the baselines make the experimental setup clearer. In addition, evaluating the method on both easier and more challenging benchmarks, comparing against the current SOTA methods, and conducting a fair comparison with the SFT+RL setting all make the paper more convincing. Therefore, I will raise my score in support of the paper.

---

### Official Review · Reviewer_VtWd · 2026-03-12

**Soundness:** 3
**Presentation:** 3
**Significance:** 3
**Originality:** 3
**Overall Recommendation:** 4
**Confidence:** 4

**Summary:**

This paper proposes StepCodeReasoner, a framework for improving code reasoning with stepwise execution supervision. The key idea is to equip code with verifiable execution anchors, train the model to predict intermediate execution states, and then apply RL with dense step-level rewards rather than relying only on the outcome reward. The training recipe combines SFT and RL, and the RL objective uses a so-called bi-level GRPO design with both inter-trajectory advantage estimation and an intra-trajectory shaping term that gives more credit to intermediate steps with more future return. The paper evaluates code reasoning benchmarks such as CRUXEval, LiveCodeBench, and REval, showing promising empirical performance, and also reports transfer to code generation benchmarks, including HumanEval, MBPP, and LiveCodeBench.

**Compliance With Llm Reviewing Policy:**

Affirmed.

**Final Justification:**

This work proposes a technically sound method for the code reasoning domain, which is important but relatively narrow. Given the limited scope in which the proposed method can be extended, I recommend accepting this work with a weak accept rating.

**Key Questions For Authors:**

- How is the Group-Relative Stepwise Advantage computed when sampled trajectories have different numbers of intermediate steps?
- How broadly do the authors expect the framework to extend beyond code reasoning?
- RL Tango [1] utilizes a step-level advantage computed across all steps from all samples within a group. Can this trick also apply to this approach? This is aksi relevant to my Question-1.

### Reference
[1] Zha et. al., RL Tango: Reinforcing Generator and Verifier Together for Language Reasoning, NeurIPS 2025.

**Limitations:**

yes

**Strengths And Weaknesses:**

## Strengths
- **The problem is important.** Sparse reward and coarse outcome supervision are real challenges for RL-based reasoning in general, and code reasoning is a natural domain where this limitation is especially visible.
- **The proposed method is technically reasonable and well-motivated.** Adding execution checkpoints and supervising intermediate states follows how humans often debug and reason about code. This makes the method intuitive and well-motivated.
- **The RL algorithm, while heuristic, shows empirical benefits.** The use of dense stepwise rewards, plus an additional intra-trajectory shaping signal to favor intermediate steps that are followed by more correct subsequent execution, is heuristic but reasonable in code reasoning tasks.
- **The empirical performance is promising.** The reported gains on code reasoning benchmarks are strong. StepCodeReasoner-7B outperforms larger baselines on the reported averages. The ablation study also supports the value of stepwise rewards beyond SFT or outcome-only RL training. The paper shows that the proposed method also improves end-to-end code generation benchmarks.

## Weakness
- **The domain of applicability is narrow.** The method depends heavily on the ability to insert and verify intermediate execution checkpoints. This is a natural fit for code reasoning, but much less so for general reasoning, open-ended generation, or domains where intermediate states are ambiguous or hard to verify. Even within coding, the core design is more directly aligned with code reasoning/execution understanding than with unconstrained code generation.
- **The RL algorithm is heuristic and may require tuning.** The intra-trajectory advantage is not a standard return estimator like GAE or accumulated return; it is a shaped heuristic that rewards a correct step more if future steps are also correct. This is intuitively reasonable, but it is still a heuristic choice.
- **The term “bi-level” is potentially confusing (minor).** The paper uses “bi-level” to describe two levels of credit assignment, but this naming can easily be confused with bilevel optimization.

---

> ### Author Rebuttal · Authors · 2026-03-30
>
> We thank the reviewer for the constructive feedback. We address each concern below.
>
> ## Weakness 1: The domain of applicability is narrow
>
> Thanks for your comment. Code reasoning is the primary focus of our work, consistent with many recent efforts: CodeI/O (Li et al., ICML 2025) shows reasoning from code transfers to broader tasks; CoRe (Xie et al., NeurIPS 2025) and CodeCrash (Lam et al., NeurIPS 2025) benchmark code semantic reasoning; rStar-Coder (Liu et al., NeurIPS 2025) uses execution-based verification for competitive programming. Most notably, Jung et al. (EMNLP 2025) demonstrate that code execution traces can serve as chain-of-thought supervision that **transfers to non-code reasoning domains**.
>
> Beyond code, our RL method is applicable to domains with verifiable intermediate states, such as agentic RL (tool call success/failure) and multi-step mathematical reasoning, which we will explore in future work.
>
> ## Weakness 2: The RL algorithm is heuristic and may require tuning
>
> We appreciate this observation. While the intra-trajectory shaping advantage (Eq. 11) is not a standard return estimator like GAE, it has a principled interpretation.
>
> The intra-trajectory advantage can be viewed as a **potential-based reward shaping** signal (Ng et al., 1999). The shaping term $r_{i,g} \cdot \frac{1}{n-i}\sum_{j=i+1}^{n} r_{j,g}$ computes the product of the current step's correctness and the empirical future success rate — analogous to weighting each step's gradient by a Monte Carlo estimate of expected future return, restricted to binary rewards. A correct step receives credit proportional to the fraction of subsequent correct steps, which is a form of undiscounted empirical Q-value estimation.
>
> Compared to GAE or full Monte Carlo returns, our formulation has two practical advantages: (1) it requires no learned value function, avoiding critic training instability; and (2) it introduces no hyperparameters beyond $\lambda$, which is robust at 0.3 across all experiments without task-specific tuning.
>
> To address the reviewer's concern, we have added this theoretical discussion to the revised paper to better motivate the design.
>
> ## Weakness 3: The term "bi-level" is potentially confusing
>
> We agree that "bi-level" may cause confusion with bilevel optimization (e.g., Stackelberg games, meta-learning). To address this, we have renamed the method to **"Dual-Granularity GRPO"** (DG-GRPO), which more accurately reflects the core idea: credit assignment at two complementary granularities — inter-trajectory (group-relative comparison) and intra-trajectory (sequential dependency shaping). We have updated the terminology throughout the revised paper.
>
> ## Question 1: How is the Group-Relative Stepwise Advantage computed when sampled trajectories have different numbers of intermediate steps?
>
> In our framework, the number of intermediate anchors is determined by the **instrumented program** $P'$, not by the model's generation. Since all $G$ trajectories in a group are sampled for the same problem context $x$ (same instrumented code), every trajectory shares the same set of $n$ execution anchors. Therefore, all trajectories within a group have identical intermediate steps, and the group-relative normalization in Eq. (10) is well-defined at each anchor index $i$.
>
> When a model fails to generate a prediction for a particular anchor (e.g., format errors or early truncation), we assign a reward of 0 for all missing anchors and the final answer, treating the trajectory as fully incorrect from that point onward.
>
> ## Question 2: How broadly do the authors expect the framework to extend beyond code reasoning?
>
> As discussed in Weakness 1, our primary focus is code reasoning, where deterministic program execution provides naturally verifiable intermediate states. The framework is also applicable to domains with verifiable intermediate states, such as agentic RL (tool call success/failure), mathematical reasoning (symbolic verification of intermediate derivations), and multi-hop QA (sub-answer evaluation). We consider these extensions a promising direction for future work.
>
> ## Question 3: Can the step-level advantage from RL Tango apply to this approach?
>
> Thank you for pointing out RL Tango. RL Tango computes a step-level advantage by pooling all steps from all samples within a group, providing a more global normalization signal. In our design, the group-relative advantage (Eq. 10) normalizes at each anchor index independently across trajectories, preserving semantic alignment (step $i$ always compared with step $i$ across trajectories). Adopting pooled normalization (as in RL Tango) could improve variance reduction but would lose this anchor-aligned comparison property.
>
> We believe the two approaches are complementary — one could apply RL Tango's pooled normalization while retaining our intra-trajectory shaping signal. To address the reviewer's question, we have added this discussion to the revised paper.

---

> > ### Author Rebuttal · Reviewer_VtWd · 2026-04-02
> >
> > Most of my concerns have been resolved, and I will keep my current positive rating. Given my evaluation of the level of novelty and the level of significance of this work, I do not increase the score further.

---

### Decision · Program_Chairs · 2026-04-30

**Decision:**

Accept (regular)

**Comment:**

This paper presents StepCodeReasoner, a framework that takes into account both intermediate program execution states as well as the final outputs for improving code reasoning and avoiding reward hacking when training only on final outputs. The paper introduces a Bi-level GRPO training algorithm (later renamed to "Dual-Granularity GRPO" (DG-GRPO)) that provides fine-grained supervision over both intermediate execution steps and final outputs. The approach is evaluated over several benchmarks including CRUXEval, LiveCodeBench, and REval, and shows significant improvements over baseline models. All reviewers found the overall approach to be interesting and novel, and appreciated the extensive experimentation showing good gains over baseline approaches. Although there were some concerns regarding generalizability and applicability of the approach to other domains, complexity of the programs that can be supported, comparisons of SFT and RL baselines with the same budget, evaluation of intermediate signals and computational overhead, scaling to larger models, generalization beyond instrumentation, and missing PRM baseline. The rebuttal response helped resolve a lot of these concerns, and it would be great to add the new experiments and discussions in the final version of the paper.